# ADAM17-dependent signaling is required for oncogenic human papillomavirus entry platform assembly

Snježana Mikuličić[1], Jérôme Finke[2], Fatima Boukhallouk[3], Elena Wüstenhagen[1], Dominik Sons[2], Yahya Homsi[2], Karina Reiss[4], Thorsten Lang[2], Luise Florin[1]*

[1]Institute for Virology and Research Center for Immunotherapy (FZI), University Medical Center of the Johannes Gutenberg-University Mainz, Mainz, Germany; [2]Department of Membrane Biochemistry, Life and Medical Sciences Institute (LIMES), University of Bonn, Bonn, Germany; [3]Institute for Medical Microbiology and Hygiene, University Medical Center of the Johannes Gutenberg University Mainz, Mainz, Germany; [4]Department of Dermatology and Allergology, University Hospital Schleswig-Holstein, Kiel, Germany

**Abstract** Oncogenic human papillomaviruses (HPV) are small DNA viruses that infect keratinocytes. After HPV binding to cell surface receptors, a cascade of molecular interactions mediates the infectious cellular internalization of virus particles. Aside from the virus itself, important molecular players involved in virus entry include the tetraspanin CD151 and the epidermal growth factor receptor (EGFR). To date, it is unknown how these components are coordinated in space and time. Here, we studied plasma membrane dynamics of CD151 and EGFR and the HPV16 capsid during the early phase of infection. We find that the proteinase ADAM17 activates the extracellular signal-regulated kinases (ERK1/2) pathway by the shedding of growth factors which triggers the formation of an endocytic entry platform. Infectious endocytic entry platforms carrying virus particles consist of two-fold larger CD151 domains containing the EGFR. Our finding clearly dissects initial virus binding from ADAM17-dependent assembly of a HPV/CD151/EGFR entry platform.
DOI: https://doi.org/10.7554/eLife.44345.001

*For correspondence:
lflorin@uni-mainz.de

**Competing interests:** The authors declare that no competing interests exist.

## Introduction

Viral infections by human papillomaviruses (HPVs) cause benign warts and malignant tumors. The small, non-enveloped virus is constituted of a circular DNA genome surrounded by a capsid mainly built of the L1 protein. More than 240 papillomavirus types have been characterized in diverse hosts, including mammals, birds and reptiles. For instance, the oncogenic HPV types 16, 18, and 31 are responsible for severe human cancers, including cervical cancer and anogenital, head and neck tumors (*Doorbar et al., 2012*).

HPV infections require a micro-wound enabling virion binding to mitotically active basal keratinocytes (*Doorbar et al., 2012*; *Ozbun, 2019*). However, virus internalization is asynchronous and slow, with a halftime of several hours (*Sapp and Bienkowska-Haba, 2009*; *Becker et al., 2018*; *Schelhaas et al., 2012*). The long halftime points toward a complex cascade of events at the cell membrane prior to virus entry into host cells. It is assumed that the sequence is initiated upon virus binding to a primary receptor, from which the virus is released in a modified form, followed by re-binding to a secondary receptor complex, which is then endocytosed (*Ozbun, 2019*; *Raff et al., 2013*; *Mikuličić and Florin, 2019*).

To date, the requirements for virus association with the secondary entry receptor and the composition of the receptor complex are not well understood. One of its main components is the tetraspanin CD151. It localizes to the cell surface, enriched in domains that associate with HPV16 particles (*Spoden et al., 2008*; *Scheffer et al., 2013*). CD151 is essential for viral endocytosis (*Spoden et al., 2008*; *Scheffer et al., 2013*; *Spoden et al., 2013*; *Scheffer et al., 2014*; *Fast et al., 2018*) and co-internalizes with the virus via a clathrin-, caveolin-, and dynamin-independent endocytic pathway (*Spoden et al., 2008*; *Spoden et al., 2013*). Integrin complexes (α6β1/β4) (*Scheffer et al., 2013*; *Evander et al., 1997*; *McMillan et al., 1999*), the annexin A2 heterotetramer (*Woodham et al., 2012*; *Dziduszko and Ozbun, 2013*), the cytoskeletal adaptor obscurin-like one protein (*Wüstenhagen et al., 2016*), and the tetraspanin CD63 (*Spoden et al., 2008*) are further components of the entry receptor complex. In addition, growth factor receptors (GFR), in particular the epidermal growth factor receptor (EGFR), were proposed as entry receptors for HPV16 (*Surviladze et al., 2012*) and multiple other viruses such as vaccinia, cytomegalo, herpes simplex, influenza, and hepatitis C virus (*Schäfer et al., 2015*). Moreover, signal transduction like the activation of EGFR and ERK signaling is a prerequisite for HPV infection (*Schelhaas et al., 2012*; *Surviladze et al., 2012*).

The activation of signaling pathways indicates that only cells in a defined activation state are susceptible for infection. What type of molecules may trigger that state? Is there a link between signaling and the entry complex?

Recent data indicate that metalloproteases are involved early in HPV16 infection (*Surviladze et al., 2012*). From the many known metalloproteases, members of "A Disintegrin And Metalloprotease" (ADAMs) family regulate multiple cell signaling pathways by the release of bioactive proteins like GFR ligands (*Gooz, 2010*; *Sahin et al., 2004*; *Horiuchi et al., 2007*). Thereby, ADAMs may represent a connection between cell signaling and components of the viral entry complex. Here, we set out to study which ADAM family member may play a role and how it may regulate the cascade of the infection pathway.

## Results

### A strong requirement of ADAM17 for the infection with oncogenic HPV pseudoviruses

Recently, it was suggested that a metalloprotease mediates early steps in HPV16 infection (*Surviladze et al., 2012*). In that study, the broad-spectrum inhibitors did not differentiate between matrix metalloproteinases (MMPs) and ADAMs (*Parvathy et al., 1998*; *Mustafi et al., 2017*; *Amour et al., 1998*; *de Meijer et al., 2011*; *Rasmussen and McCann, 1997*; *Murphy, 2011*). From the ADAM family, ADAM10 and ADAM17 are the most likely candidates as they are highly expressed in keratinocytes (*Lee et al., 2011*; *Maretzky et al., 2008*; *Sommer et al., 2016*; *Brooke et al., 2014*; *Hirayama et al., 2017*) and involved in the shedding of numerous substrates (*Reiss and Saftig, 2009*; *Edwards et al., 2008*).

Infecting HaCaT keratinocytes with HPV16 pseudovirions (PsVs), we applied the broad-spectrum metalloproteinase inhibitor TAPI-0, the preferential ADAM10 inhibitor GI254023X (GI) and the mixed ADAM10/ADAM17 inhibitor GW280264X (GW) (*Figure 1—figure supplement 1A*). TAPI-0 and GW diminished infection, whereas the ADAM10 inhibitor GI showed no significant effect. This is pointing toward a requirement of ADAMs, in particular of ADAM17 rather than ADAM10. To differentiate between these two ADAMs with a more specific approach, ADAM17 or ADAM10 were depleted by siRNA knockdown. For ADAM10 and ADAM17, respectively, two different siRNAs were employed in three epithelial cell models, including HeLa cells, HaCaT cells, and primary normal human epithelial keratinocytes (NHEK). Under all conditions ADAM17 depletion consistently and strongly decreased infection (*Figure 1A–C*). In contrast, ADAM10 depletion showed no clear effect (*Figure 1—figure supplement 1B–D*). ADAM17 was also required for the infection by two other highly oncogenic HPV types, HPV18 and HPV31 (*Figure 1D*). Altogether, this indicates that ADAM17 is important for the infection with oncogenic human papillomaviruses.

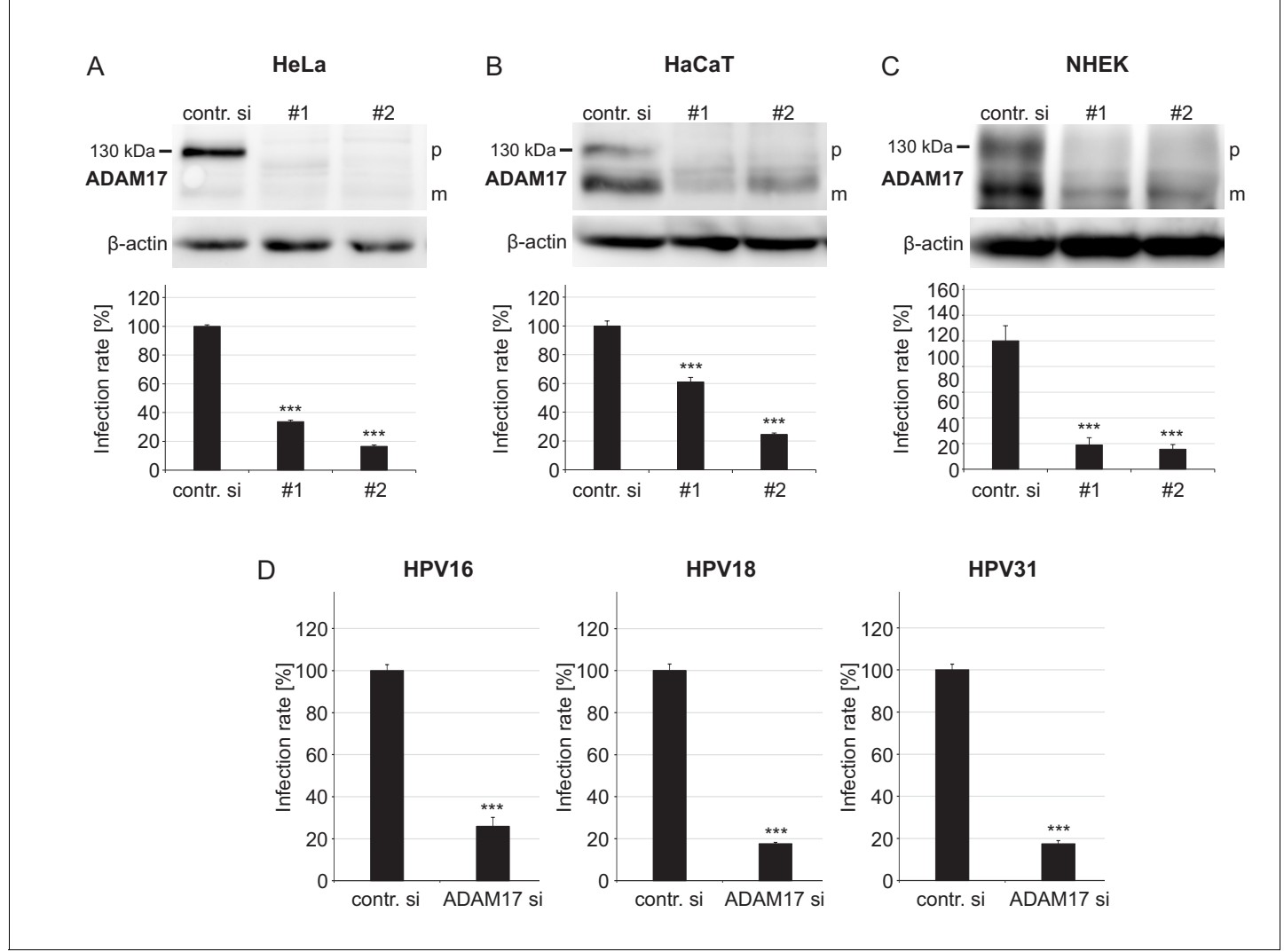

**Figure 1.** ADAM17 is required for HPV infection. (**A**) HeLa, (**B**) HaCaT and (**C**) NHEK cells were treated with each one of two specific ADAM17 siRNAs and 48 hr later incubated with PsVs. Data obtained from HeLa cells (n = 9–10) were analyzed with two sample t-test: p=1.77E-12 (#1) and Welch two sample t-test: p=1.62E-09 (#2). HaCaT (n = 10); two-sample t-test: p=1.23E-07 (#1) and Welch two sample t-test: p=5.34E-10 (#2). NHEK (n = 8–13); Welch two sample t-test: p=1.07E-05 (#1) and p=7.35E-06 (#2). Upper panels, immunoblots for ADAM17 illustrating the efficiency of ADAM17 depletion (β-actin served as a loading control), whereas lower panels display the infection rate. Abbreviations: p and m indicate the premature and mature form of ADAM17, respectively. (**D**) HaCaT cells were depleted of ADAM17 applying a mixture of the two different pooled ADAM17-specific siRNAs (ADAM si) and 48 hr later infected with HPV16, HPV18 or HPV31 PsVs. Data for HPV16 (n = 9–10) and HPV31 (n = 10) were analyzed with two sample t-test: p=3.54E-11 and p=9.11E-16, respectively. Data for HPV18 (n = 10) with Welch two sample t-test: p=2.17E-10. (A)-(D) The values are given as mean ± SEM and the mean for control siRNA-treated cells (contr. si) was set to 100%. ADAM17-specific siRNAs have no effect on luciferase expression (*Figure 1—figure supplement 2*).

DOI: https://doi.org/10.7554/eLife.44345.002

The following figure supplements are available for figure 1:

**Figure supplement 1.** Effect of ADAM17 inhibitors and ADAM10 depletion on HPV16-mediated infection rate.

DOI: https://doi.org/10.7554/eLife.44345.003

**Figure supplement 2.** ADAM17-specific siRNAs have no effect on luciferase expression.

DOI: https://doi.org/10.7554/eLife.44345.004

# ADAM17 activity triggers the formation of a HPV16/CD151/EGFR platform

The tetraspanin CD151 defines the entry receptor complex (*Scheffer et al., 2013*; *Scheffer et al., 2014*). Therefore, we asked whether ADAM17 is required for the association of HPV16 with CD151. Prior to testing the role of ADAM17, we characterized the time course of virus colocalization with the entry receptor component CD151. HaCaTs were incubated with PsVs for 15 min, washed, and fixed or incubated up to several hours prior to fixation. Directly after virus incubation (marked as time point 0:00), we found virus particles associated with the cell surface, but they poorly overlapped with CD151-enriched domains (*Figure 2—figure supplement 1*). One hour after adding PsVs (corresponds to time point 0:45), there was a moderate increase in CD151-L1 overlap that further increased in the next 2 hr, apparently plateauing after about 5 hr (*Figure 2—figure supplement 1*). This is in line with our previous time course analyses of the CD151-L1 interaction on HeLa cells (*Spoden et al., 2008*). This slow association of viral particles with CD151-enriched domains implies the idea of a slowly maturing HPV16 entry platform.

To learn more about this process, we analyzed the CD151-enriched domains employing nanoscopy as previously done for other tetraspanins (*Homsi et al., 2014*; *Zuidscherwoude et al., 2015*). Superresolution STED microscopy revealed a defined nanocluster pattern suggesting that larger CD151-enriched domains, as recorded by diffraction-limited microscopy, are in fact closely associated CD151 nanoclusters. The resolved nanocluster pattern enabled us to determine the distance between a bound viral particle to its next nearest CD151 nanocluster (*Figure 2A*). The average size of a CD151 cluster is roughly 200 nm (see Figure 4), and an HPV particle has a size of 55 nm (*Baker et al., 1991*). Therefore, distances shorter than 150 nm allow for physical contact between a viral particle and a CD151 nanocluster. Interestingly, we uncovered a large fraction of PsVs in close proximity to CD151 nano-clusters already directly after virus binding. The fraction of such closely associated viral particles strongly increases over time (*Figure 2A and B*). We cannot exclude that we underestimate the increase of this fraction due to preferential internalization of tight CD151-viral particle associates. Importantly, as well the CD151 brightness at the site of viral particle binding increases (*Figure 2C*), suggesting that the viral particle enriches CD151 in preparation of endocytosis.

Enrichment of CD151 at viral particle binding sites can be due to viral particles moving toward CD151 clusters, stationary viral particles trapping CD151, or a mixed scenario. With TIRF microscopy, we studied the dynamics of viral particles and CD151. While viral particles were clearly immobile, CD151 moved fast (*Figure 2—figure supplement 2*). These observations are in line with previous reports showing HPV16 PsV confinement on the cell body (*Schelhaas et al., 2008*) and that CD81 tetraspanin clusters are stable only for a few seconds (*Homsi et al., 2014*). We cannot exclude that in our CD151 tracking study, in which diffraction limited microscopy did not well resolve individual CD151 nanoclusters, we monitor larger intracellular vesicles. However, if viral particles are stationary and tetraspanins are mobile (*Espenel et al., 2008*), platform maturation presumably involves CD151 nucleation at viral particle binding sites. The increase in CD151 intensity at PsVs binding sites (*Figure 2C*) illustrates this is a slow process that can take up to several hours.

Next, we tested whether ADAM17 plays a role in the association of viral particles and CD151-enriched domains in two different epithelial cell lines. 5 hr after initial PsVs contact, we found a significant effect. ADAM17 depletion reduced virus-CD151 overlap by roughly a third in HaCaTs or NHEK (*Figure 2D–F*). The data reveals ADAM17-dependent spatial proximity between virus and CD151. Still, it is possible that viral particles and CD151 domains, although overlapping in the micrographs and exhibiting distances to each other below 150 nm, do not physically interact with each other. To corroborate spatial proximity with a better resolution, we turned to the Duolink proximity ligation assay (PLA) (*Fredriksson et al., 2002*; *Söderberg et al., 2006*). In the PLA bright fluorescent spots form at sites where two different molecular entities are closer than $\approx$ 40 nm (*Bagchi et al., 2015*; *Mendez and Banerjee, 2017*). Hence, in this assay fluorescent spots indicate sites at which viral particles and CD151 domains should indeed be close enough to interact physically. In the PLA, we found that the PLA signal was strongly dependent on ADAM17 (*Figure 2G and H*). Thus, the microscopic analysis and the PLA suggest that ADAM17 mediates close association between HPVs and CD151-enriched domains. This means that ADAM17 either is required for initial virus binding, or

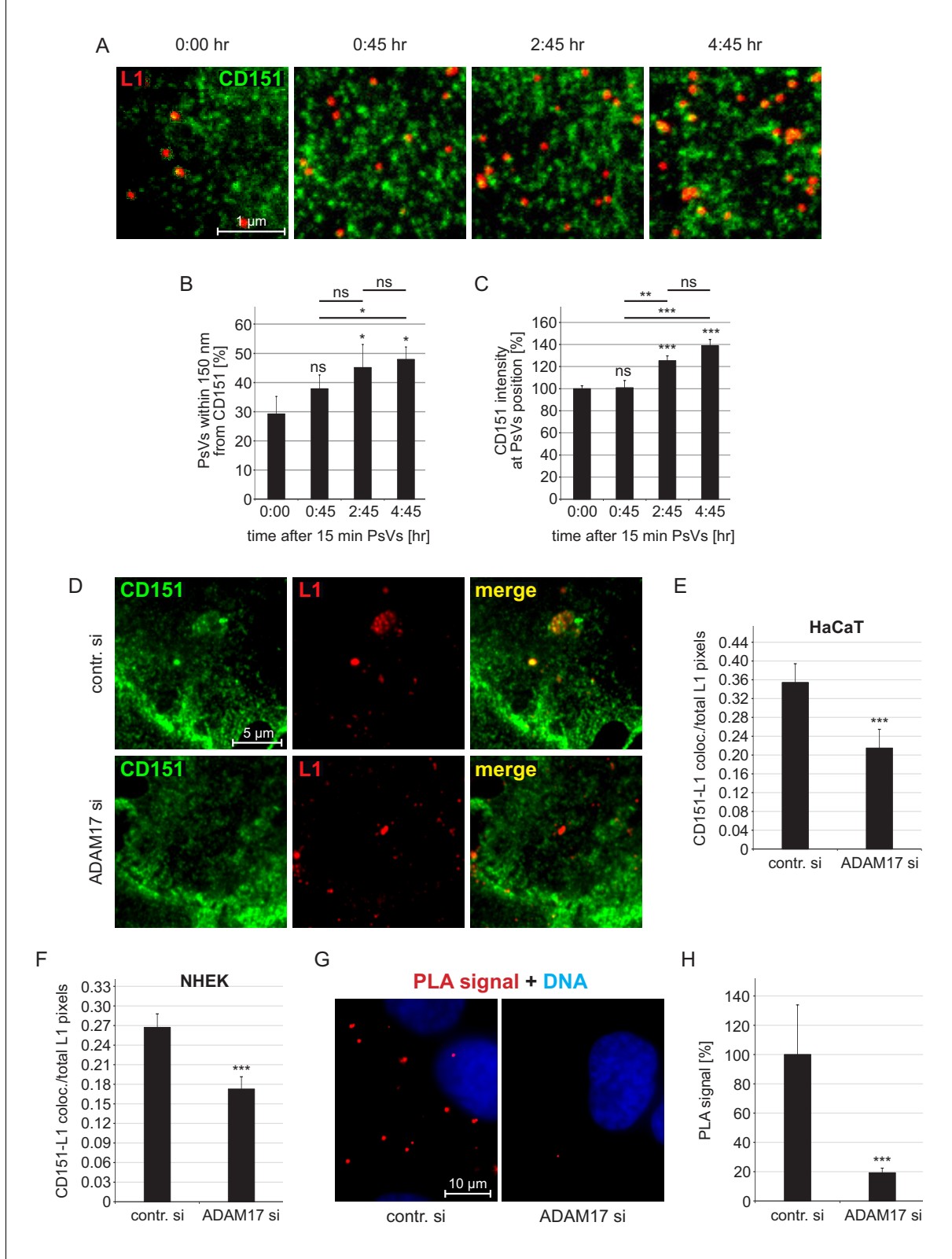

**Figure 2.** Association of HPV16 with CD151. (**A**) HaCaT cells were incubated for 15 min with HPV16 PsVs, washed, and either directly fixed or after incubation for the indicated times. Then cells were stained with anti-CD151 mAb 11G5A (green) and anti-L1 pAb K75 (red) antibodies and analyzed by superresolution STED microscopy. Representative STED-images for the different time points are shown. From the STED micrographs, we determined the distance of a PsV to its next nearest CD151 cluster and the local intensity of CD151 at the PsV position. (**B**) Increase of the fraction of HPV16 PsVs

*Figure 2 continued on next page*

*Figure 2 continued*

that are closer than 150 nm to the next CD151 nano-cluster. Statistical significance (n=3) was analyzed with paired t-test: p=0.091 (0:45), p=0.021 (2:45), p=0.017 (4:45), difference between 0:45 and 2:45: p=0.276, between 0:45 and 4:45: p=0.011 and 2:45 and 4:45: p=0.617. Values for every sample were expressed as percentage of the total number of PsVs in the given sample. (C) The CD151 intensity at the HPV16 PsV position increases over time. Data (60–65 cells per condition) were analyzed with Wilcoxon rank sum test: p=0.974 (0:45), p=2.53E-08 (4:45), difference between 0:45 and 2:45: p=4.86E-03, between 0:45 and 4:45: p=3.02E-04 and 2:45 and 4:45: p=0.290, and with Welch two sample t-test: p=5.20E-06 (2:45). 15 min time point (0:00) was normalized to 100%. (D) HaCaT cells were incubated for 15 min with PsVs, washed, incubated for another 4:45 hr, fixed and incubated with primary antibodies as in (A). Representative images are shown of control siRNA- (upper panel) or ADAM17 siRNA- (lower panel) treated cells. (E, F) Quantification of the spatial overlap between CD151 and L1 in HaCaTs (E) or in NHEK cells (F). The values obtained from the ratio of CD151-L1 colocalizing to total L1 pixels of control siRNA-treated cells (contr. si) were normalized to 100%. Data for HaCaTs (n = 111–129) were analyzed using Wilcoxon rank sum test: p=4.33E-04. NHEKs (n = 180–187); Wilcoxon rank sum test: p=1.75E-05. (G) HaCaT control or ADAM17 siRNA-treated cells were incubated with HPV16 PsVs for 5 hr, fixed, and processed for PLA detection using primary antibodies as in (A). Shown is the CD151-L1 PLA signal. (H) The ratio of PLA positive signal (red) to nuclear signal (blue) is for contr. si set to 100%. Data (n = 104–115 images) were analyzed using Wilcoxon rank sum test: p=5.77E-10. All values are given as mean ± SEM.

DOI: https://doi.org/10.7554/eLife.44345.005

The following figure supplements are available for figure 2:

**Figure supplement 1.** Time course of CD151-L1 colocalization.
DOI: https://doi.org/10.7554/eLife.44345.006
**Figure supplement 2.** Viral particle mobility analyzed by TIRF microscopy.
DOI: https://doi.org/10.7554/eLife.44345.007
**Figure supplement 3.** ADAM17 plays no role in cell surface binding of HPV16 PsVs.
DOI: https://doi.org/10.7554/eLife.44345.008
**Figure supplement 4.** ADAM17 does not cleave major capsid protein L1 nor affects HPV16 internalization rate but priming/processing of the L1 protein in an indirect manner.
DOI: https://doi.org/10.7554/eLife.44345.009

it plays a role somewhere between binding and endocytic platform formation, for example for L1 cleavage/priming by ADAM17 or another protease.

To investigate a role of ADAM17 in virus binding, we employed flow cytometry analysis testing whether ADAM17 depletion diminishes the number of PsVs on the cell surface, which was not the case (*Figure 2—figure supplement 3*). Therefore, we exclude ADAM17 requirement for initial virus binding to the cell surface. Investigating the activity of ADAM17 (using TGF-α as positive control) and the cleavage pattern of the viral L1 protein (L1 priming), we find that L1 is neither a substrate for ADAM17 (*Figure 2—figure supplement 4A and B*) nor is internalization of viral particles efficiently inhibited by ADAM17 depletion (*Figure 2—figure supplement 4C–D*). Instead, L1 cleavage products substantially decreased in ADAM17-depleted cells (*Figure 2—figure supplement 4E and F*). Therefore, we speculate that ADAM17 activity is required for viral particle association with L1-priming proteases such as kallikrein-8 prior to endocytosis and additional proteases in intracellular compartments of the infectious entry pathway (*Cerqueira et al., 2015*). Inhibition of these early steps on the plasma membrane rather leads to virus internalization into a non-infectious entry pathway as described earlier (*Selinka et al., 2007*).

Growth factor receptors like EGFR have been proposed as additional components of the HPV16 entry receptor complex (*Surviladze et al., 2012*). As anticipated, in HaCaT cells we find microscopic overlap between EGFR and L1 (*Figure 3A*), which was significantly diminished in the absence of ADAM17 (*Figure 3A and B*).

Next, we employed fluorescence microscopy with image-deconvolution studying double staining for CD151 and EGFR (*Figure 3C and D*). The similarity of the CD151 and EGFR localization patterns indicate co-enrichment of the two membrane proteins.

Moreover, superresolution STED microscopy and quantitative analysis thereof revealed some overlap of the tetraspanin with EGFR (*Figure 3D and E*) in line with previous reports demonstrating that growth factor receptors are recruited into tetraspanin-enriched microdomains (*Berditchevski and Odintsova, 2016*; *Zona et al., 2013*; *Bruening et al., 2018*). However, overlap was independent of ADAM17 (*Figure 3E*).

In conclusion, this indicates that ADAM17 is required for the association of HPV16 with the secondary entry receptor complex, which is composed of pre-formed CD151-EGFR nanocluster assemblies.

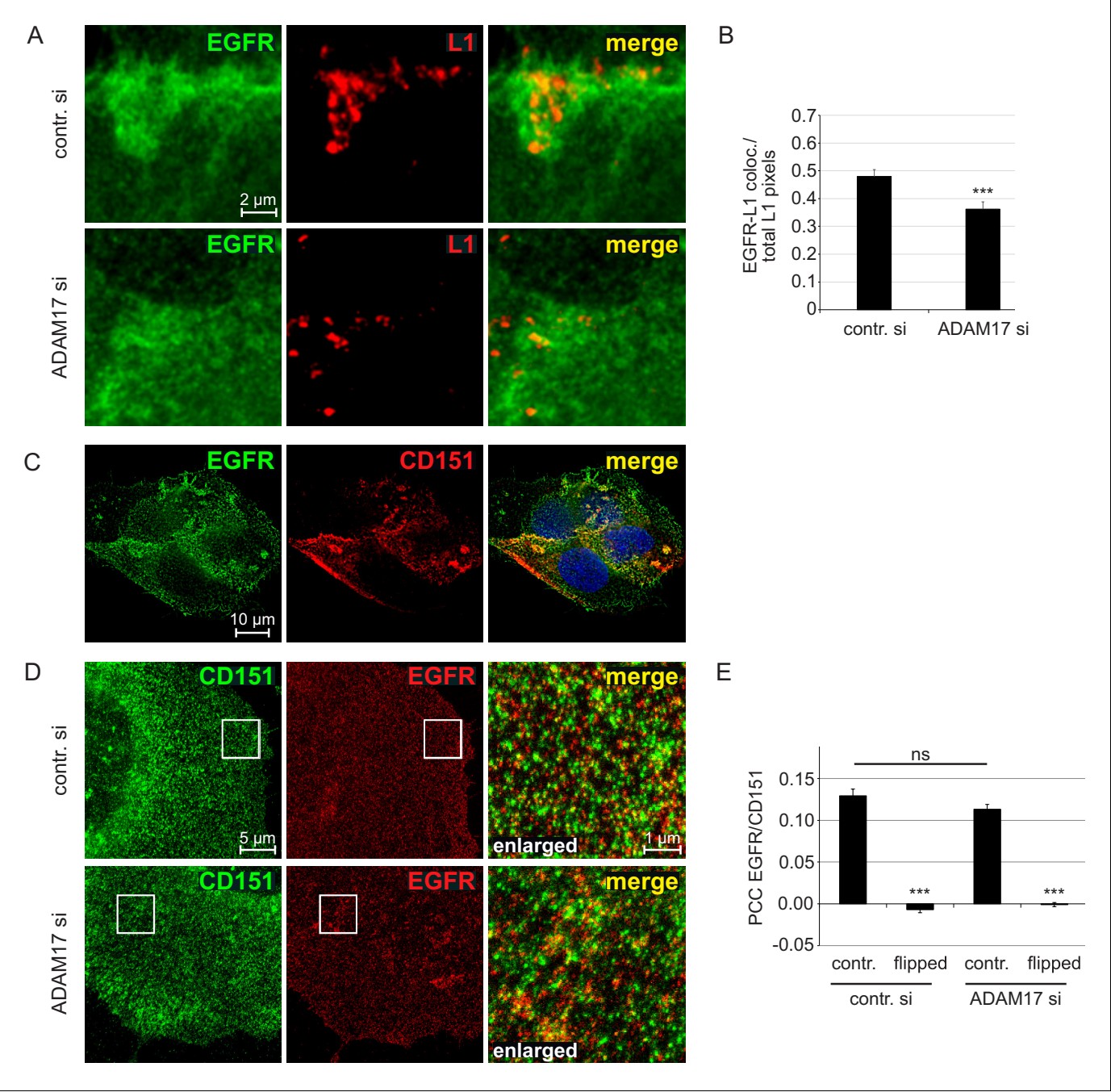

**Figure 3.** EGFR as a component of the HPV entry receptor complex. (**A, B**) ADAM17 depletion reduces EGFR-L1 association. (**A**) Representative images of HaCaT cells treated with control or ADAM17-specific siRNA for 48 hr, exposed to HPV16 PsVs for 5 hr, and subsequently stained with EGFR-specific D38B1 (green) and L1-specific 312 F (red) antibodies. (**B**) Quantification of the pixel overlap from the stainings shown in (A). Data for HaCaTs (n = 158–176 randomly chosen fields) were analyzed using Wilcoxon rank sum test: p=5.90E-04. (**C**) EGFR-CD151 association analyzed by fluorescence microscopy. Shown is a deconvoluted image of a HaCaT cells double stained with antibodies specific for EGFR (D38B1) and CD151 (11G5A). EGFR is shown in green, CD151 in red and nuclei in blue. (**D**) EGFR-CD151 overlap analyzed by superresolution microscopy. Representative images of control or ADAM17 depleted HaCaT cells, exposed to HPV16 PsVs for 5 hr, stained with the same primary antibodies as in (C) but CD151 is shown in green and EGFR in red, and analyzed by STED microscopy. (**E**) Pearson's correlation coefficient (PCC) of staining shown in (D) indicate specific overlap of EGFR and CD151. For the quantification of random overlap, we flipped one image horizontally and vertically to randomize the distribution. Data (n = 60 randomly chosen fields) for each siRNA was compared to control (contr. si) and analyzed using Wilcoxon rank sum test: p<2.20E-16. The difference between contr. and ADAM17 siRNA was not significant (p=0.239). The values for all experiments are given as mean ± SEM.
DOI: https://doi.org/10.7554/eLife.44345.010

## Endocytosis of large CD151 domains

To investigate possible changes of CD151 clusters during virus infection, we treated HaCaT cells with PsVs for 5 hr and analyzed by STED microscopy the immunostaining pattern of CD151 on membrane sheets generated prior to fixation (*Homsi et al., 2014*). This treatment ensures that only structures located at the plasma membrane are analyzed (*Figure 4—figure supplement 1*). We find that CD151-enriched domains in PsVs-non-exposed cells have an average size of $\approx$ 210 nm (*Figure 4B*). Often several individual CD151 clusters gather together to groups (*Figure 4A*). PsVs reduced the cluster diameter to $\approx$ 150 nm, which is accompanied by a $\approx$ 25% reduction of the overall CD151 staining intensity (*Figure 4B and C*). A reduction of the CD151 plasma membrane level after viral uptake is expected, as CD151 is co-internalized with viral particles (*Scheffer et al., 2013*). Reduction of cluster diameter and CD151 level both require ADAM17 (*Figure 4B and C*). Hence, the data indicate that only in the presence of ADAM17 PsVs-CD151 entities are efficiently endocytosed.

The reduction from 210 to 150 nm may suggest an only modest difference in size. However, a reduction of the cluster diameter by a quarter corresponds to a halving of the area. Moreover, reduction of the diameter in conjunction with a diminished CD151 level suggests the preferential internalization of larger CD151 clusters. Thus, viral particles induce endocytic uptake of two-fold larger than average CD151 platforms.

## ADAM17 releases bioactive compounds

Virus particles or signaling molecules could bind back to the cells from which they were released, or diffuse toward distal cells (*Figure 5A*). In the following, we examined whether the sheddase activity of ADAM17 is required for the release of virus particles from the primary binding site or alternatively, whether it produces bioactive compounds like signaling molecules that are required for platform maturation.

To study this question and to differentiate between the two steps of release and re-binding, we used a co-culture transwell assay (*Figure 5B*). The experimental setup restrains direct cell-to-cell contact, but allows exchange of soluble compounds like viruses and bioactive molecules, for example growth factors (GF), between donor and recipient cells. In this assay, cells releasing substances are called donor cells (donors). They are cultured in inserts above recipient cells (recipients) placed in the same chamber and thus able to receive substances released from donors.

It has been previously shown that donor cells support infection of recipient cells (*Surviladze et al., 2012*). In our assay, control donor or ADAM17-depleted HaCaT cells were incubated with HPV16 PsVs for 1 hr, intensively washed to remove unbound virus, placed above recipient cells, and the next day assayed for infection. As expected, ADAM17 depletion in donor cells strongly diminished their infection, but also halved the infection of non-ADAM17-depleted recipient cells (*Figure 5C*). This demonstrates that virus particles were released from donor cells and diffused to the recipient cells. However, there are several possible explanations for the lower infection rate of non-ADAM17-depleted recipient cells: recipients have received (i) less virus particles, (ii) the same amount but less infectious particles, or (iii) less bioactive molecules required for signaling pathway activation.

For clarification, we examined whether the quantity of L1 protein in the supernatant, which is an indicator for released viral particles, is ADAM17-dependent. We found no difference between control and ADAM17-depleted cells (*Figure 5D*). This suggests that the release of virus particles from primary attachment receptors does not require ADAM17, which is in line with our previous finding where the quantity of surface bound virus particles was shown to be ADAM17-independent (*Figure 2—figure supplement 3*). Thus, no difference in released L1 excludes the possibility that recipient cells receive less virus particles.

Are then virus particles less infectious when released from ADAM17-depleted cells? To test this, the co-culture assay was performed under inverted conditions. Non-treated donor cells were incubated with HPV16 PsVs to ensure the release of identical virus particles for control and ADAM17-depleted recipient cells. In the case that released viral particles require ADAM17 for being infectious, ADAM17 depletion in recipient cells should have no effect on their infection rate. However, we observed the opposite, infection in recipient cells was almost completely blocked (*Figure 5E*, lower panel). Therefore, the involvement of ADAM17 in the release of soluble bioactive compounds, as for example growth factors, which activate specific signaling pathways for infection and can

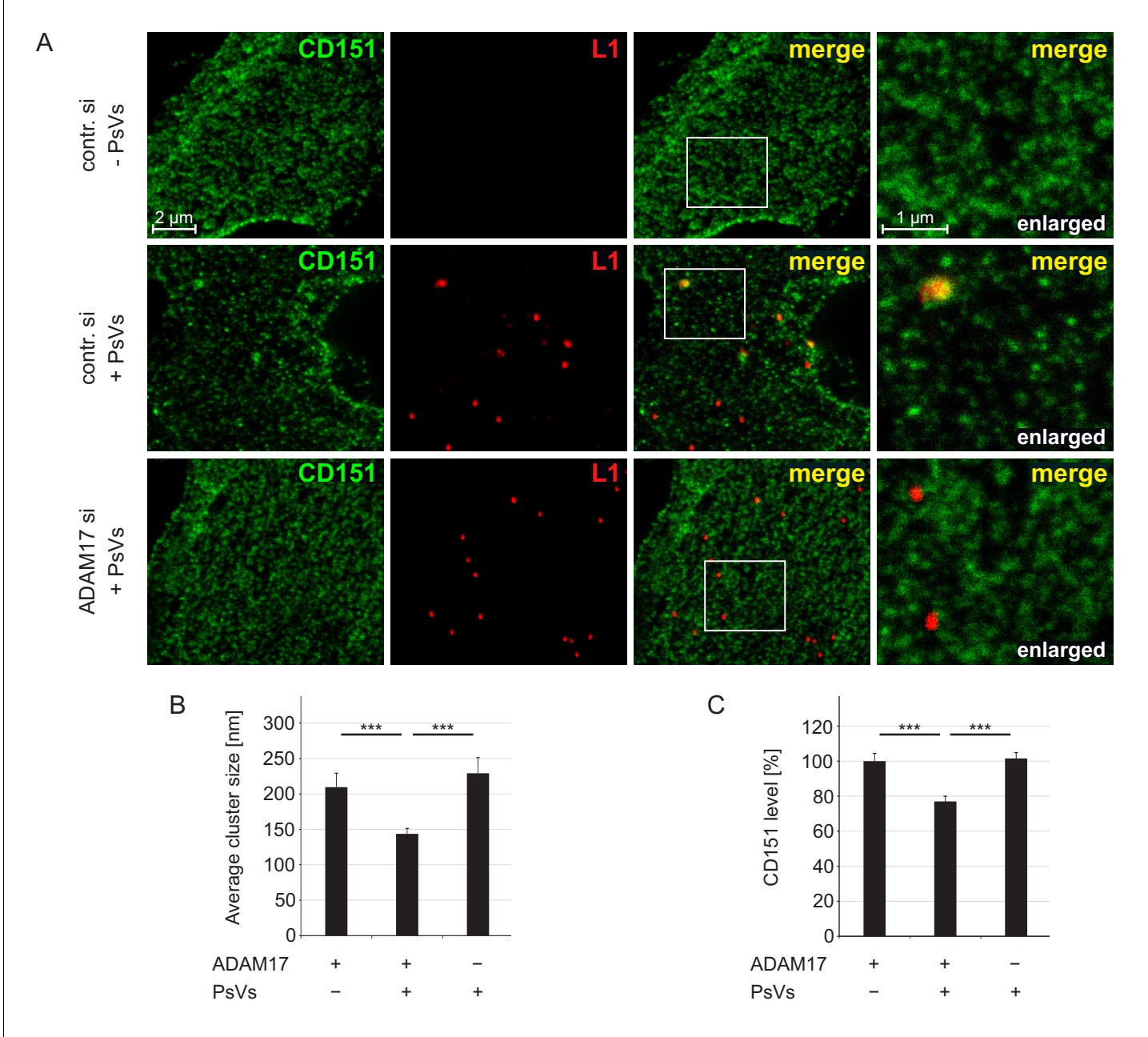

**Figure 4.** ADAM17 is required for internalization of large CD151 clusters. (**A**) Representative images of membrane sheets generated from HaCaT cells stained for CD151 with 11G5A mAb (green) and HPV16 L1 with K75 pAb (red). Shown are membrane sheets generated from control siRNA (upper and middle row) and ADAM17 siRNA (lower row) transfected cells that were incubated for 5 hr without (upper row) or with PsVs (middle and lower row). (**B**) HPV16 treatment diminishes CD151 cluster size, an effect that is abolished upon ADAM17 depletion. ADAM17+/PsVs + was compared to the two other conditions and analyzed with Wilcoxon rank sum test: p=3.10E-04 (for ADAM17+/PsVs-) and p=7.01E-09 (for ADAM17-/PsVs+). (**C**) HPV16 treatment diminishes the CD151 surface level, an effect that is abolished upon ADAM17 depletion. Statistical analysis was performed comparing to the condition ADAM17+/PsVs + and analyzed with Welch two sample t-test: p=4.16E-05 (for ADAM17+/PsVs-) and two sample t-test: p=2.21E-07 (for ADAM17-/PsVs +). For (**B**) and (**C**) (n = 60), membrane sheets for each condition were collected from three biological replicates. Values are given as mean ± SEM. The condition ADAM17+/PsVs- was set to 100%.

DOI: https://doi.org/10.7554/eLife.44345.011

The following figure supplement is available for figure 4:

**Figure supplement 1.** Illustration of the membrane sheet preparation.

DOI: https://doi.org/10.7554/eLife.44345.012

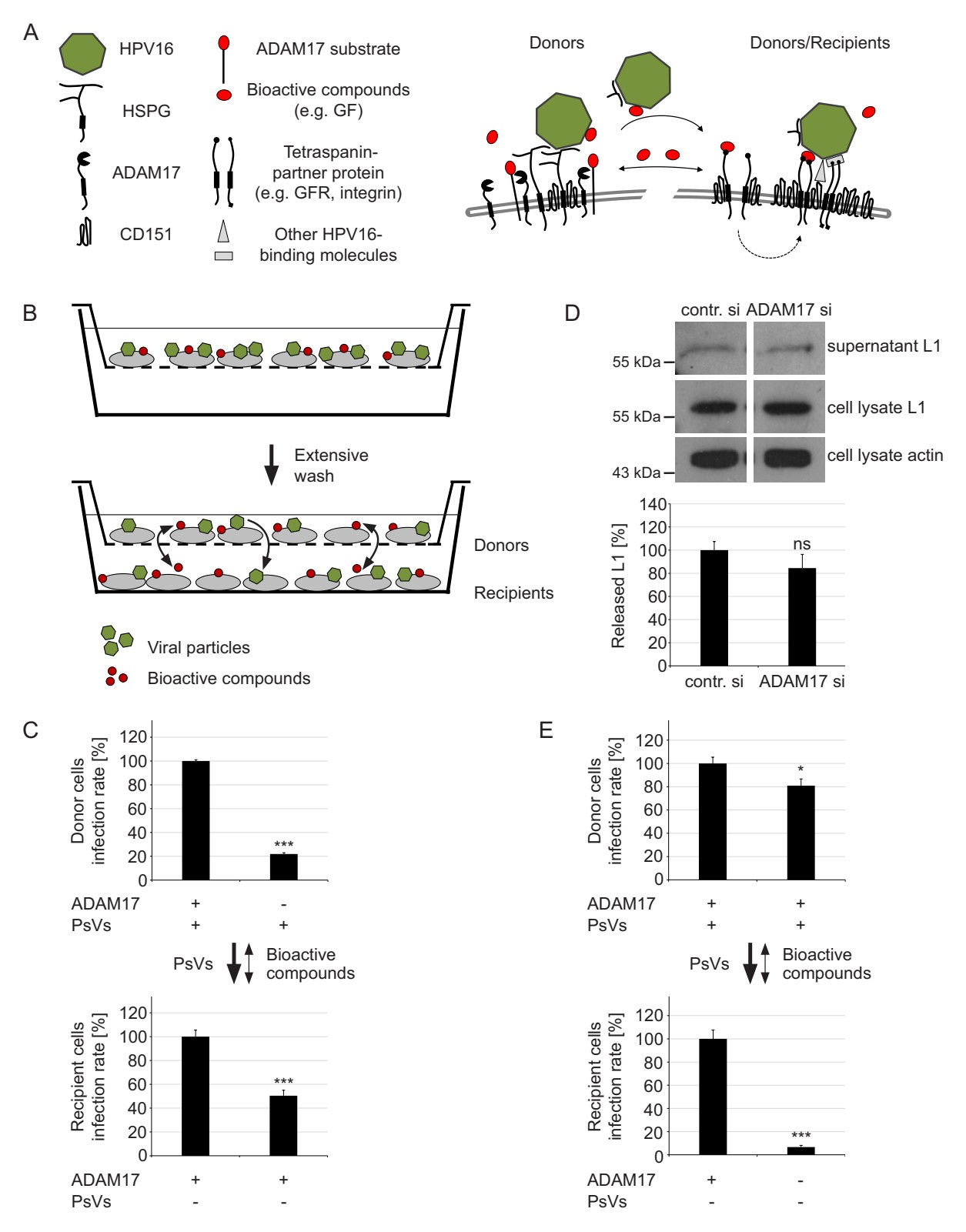

**Figure 5.** ADAM17 acts by the release of bioactive compounds. (**A**) Two-step model of virus entry platform formation. Initially, the virus (green) binds to a primary complex (left) from which it is transferred to or transformed into a secondary receptor entry complex (right). Cells exchange bioactive molecules (red) in both directions. (**B**) Schematic illustration of the "donor" cell - "recipient" cell co-culture system. Donor cells cultured in inserts were incubated with PsVs, washed and placed above recipient cells. Black arrows indicate transfer of PsVs and bioactive compounds between the donor and

*Figure 5 continued on next page*

*Figure 5 continued*

recipient cells. (C) Donor cells were transfected with control or ADAM17 siRNA and 48 hr later incubated with PsVs for 1 hr to allow virus binding. Next, cells were intensively washed, placed on top of siRNA-non-treated and PsVs-non-exposed recipients and incubated for another 24 hr. Relative infection rates obtained for donor cells (n = 11–13) were analyzed using Welch two sample t-test: p=4.81E-04 and for recipients (n = 12) with two sample t-test: p=5.34E-04. (D) HaCaT cells were depleted of ADAM17 and 48 hr later incubated with HPV16 PsVs for 15 min. Next, the cells were thoroughly washed, incubated for another 4 hr, and finally L1 was assessed by western blot using L1-specific Ab 312F (upper panel). Shown is a percentage of the L1 band intensity and represents the amount of the total L1 protein in the supernatant (lower panel). Data (n = 7–8) were analyzed using Wilcoxon rank sum test: p=0.536. Detection of L1 and β-actin in the cell lysate served as controls. The values are given as mean ± SEM and the mean for control siRNA-treated cells (contr. si) was set to 100%. (E) Recipient cells were transfected with control or ADAM17 siRNA. After 48 hr siRNA-non-treated donor cells were allowed to bind PsVs for 1 hr, intensively washed, placed on top of siRNA-treated and PsVs-non-exposed recipient cells and incubated for another 24 hr. Data for donor cells (n = 12) were analyzed with two sample t-test: p=0.025 and for recipients (n = 11–12) with Welch two sample t-test: p=1.36E-07. For (C) and (E) infection rate of donor (upper panel) and recipient (lower panel) cells is displayed as a percentage. The values are given as mean ± SEM and the mean for control siRNA-treated cells was set to 100% for donors and recipients. Symbols: + and – denote presence or absence of ADAM17 protein or PsVs, respectively.

DOI: https://doi.org/10.7554/eLife.44345.013

also decorate the virus (*Surviladze et al., 2012*), might explain these findings. Hence, bioactive molecules are required for viral infection, no matter whether the viral particle is released and re-binds to the cell or not. The hypothesis on the involvement of released growth factors is supported by the observation that the non-ADAM17-depleted donor cells in the inverted assay configuration are significantly less infected when the corresponding recipient cells are ADAM17-depleted (*Figure 5E*, upper panel). In contrast to one-directional diffusion of virus particles, bioactive compounds are exchanged in both directions, meaning that the donor cells sense less bioactive compounds when the recipient cells are ADAM17-depleted. However, the effect is small, suggesting that biomolecules rather act locally and do not diffuse over large distances. We further speculate that, although viral particles are released, they relatively fast rebind to cell surface receptors, explaining the large fraction of virus particles binding close to CD151 clusters already after 15 min (*Figure 2B*).

## ADAM17 triggers virus entry platform formation via the ERK signaling pathway

What may be the type of released biomolecules or signaling pathway? The literature suggests a link between ADAM17 and infection pathways regulated via the mitogen-activated protein kinase (MAPK) ERK1/2 signaling pathway because HPV16 infection requires activation of ERK signaling via EGFR (*Surviladze et al., 2012*) and ADAM17 mediates cleavage of growth factors that activate ERK1/2 (*Kansra et al., 2004*; *Göoz et al., 2006*; *Sommer et al., 2016*). Thus, we used the EGFR blocking-antibody Cetuximab to confirm the functional requirement of EGFR in HPV16 PsVs infection (*Figure 6A*). Next, we tested whether ADAM17 is required for ERK1/2 phosphorylation in our system (*Figure 6B and C*, and *Figure 6—figure supplement 1A and B*). We found that in HaCaT cells ADAM17 depletion reduces ERK1/2 phosphorylation, independent from the treatment with PsVs.

To solidify the hypothesis that ADAM17 acts via the ERK signaling pathway in HPV16 infection and entry platform formation, we restored ERK activation by the addition of recombinant soluble form of the epidermal growth factor (EGF) which is one of the EGFR-activating ligands. Supplying the medium with EGF did not only abolish diminished ERK phosphorylation after ADAM17 depletion (*Figure 6A and B*), but as well restored infection in ADAM17-depleted cells (*Figure 6D*). Moreover, EGF recovered the diminished CD151/L1 PLA signal in ADAM17-depleted cells (*Figure 6E and F*). These findings show that soluble biomolecules activate the ERK1/2 signaling pathway, a prerequisite for HPV16 entry platform formation and infection.

## ADAM17-dependent GF shedding links the ADAM17 activity to ERK signaling and HPV16 infection

To uncover the identity of a specific ADAM17 substrate which unambiguously links the sheddase to the ERK signaling pathway and virus infection, we analyzed the release of amphiregulin (AREG), the transforming growth factor alpha (TGF-α), and heparin-binding EGF-like growth factor (HB-EGF) into the supernatant of HaCaT cells and their functional requirement in HPV16 infection (*Figure 7*). The pro-forms of these GFs are well-known human ADAM17 substrates (*Gooz, 2010*; *Sahin et al., 2004*;

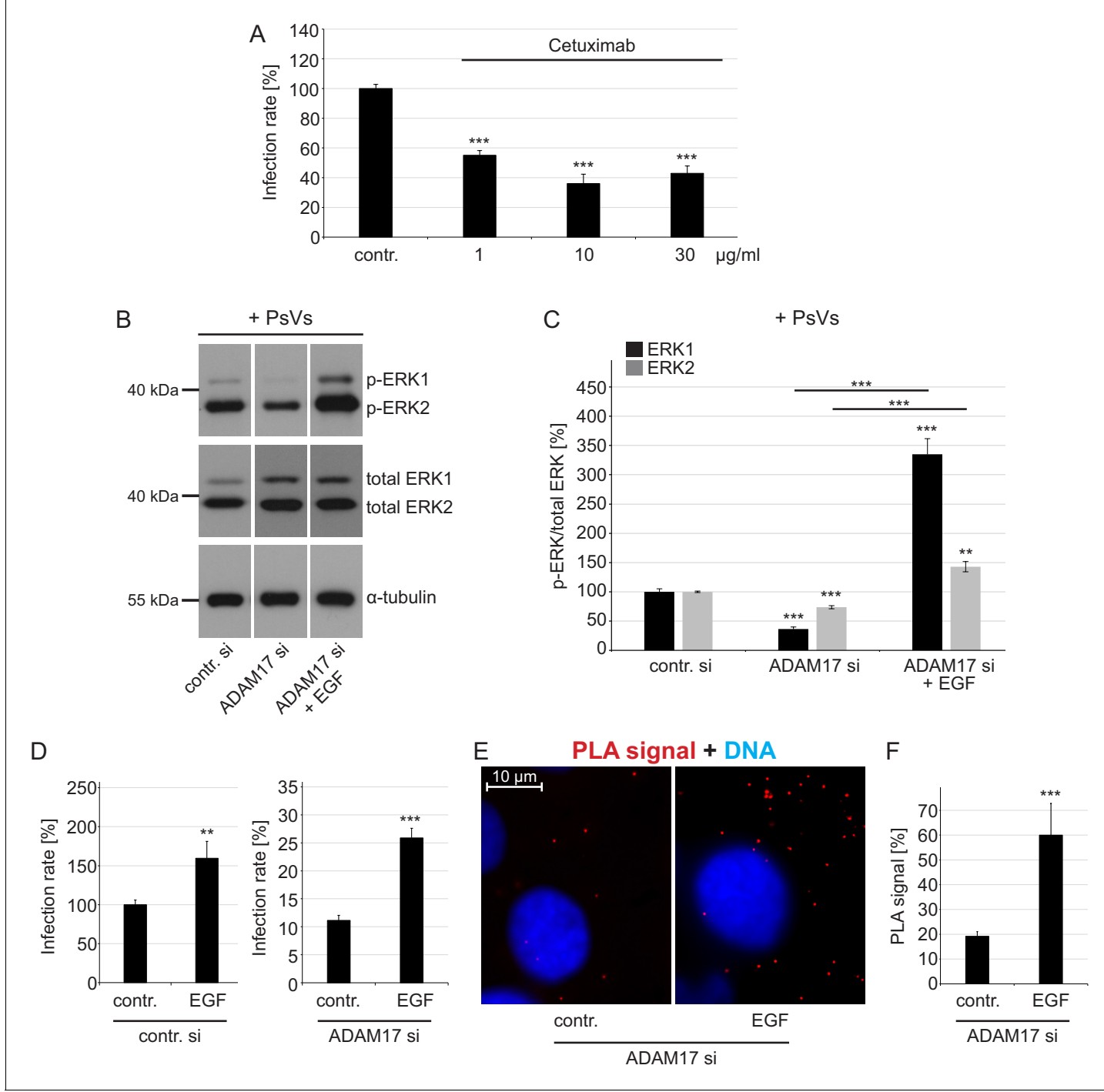

**Figure 6.** EGFR is a proviral factor in HPV16 entry (**A**) and soluble EGF relieves the effect of ADAM17 depletion on the phosphorylation status of ERK1/ 2 (**B, C**), HPV16 PsVs infection rate (**D**) and HPV16-CD151 proximity (**E, F**). (**A**) EGFR blocking-antibody Cetuximab decreases HPV16 PsVs infection rate. HaCaT cells were pre-incubated with stated concentrations (μg/ml) of Cetuximab for 1 hr, infected with HPV16 PsVs and 24 hr later luciferase counts were measured. Samples (n = 8) were analyzed using two sample t-test: p=3.67E-08 (1 μg/ml), and Welch two sample t-test: p=4.42E-06 (10 μg/ml) and p=6.76E-07 (30 μg/ml). Infection rate is given as mean ± SEM and the mean for control-treated cells (contr.) was set to 100%. (**B**) HaCaT cells were depleted of ADAM17 for 48 hr. Next, cells were starved for 1 hr in medium without FCS, either left non-treated or treated with EGF (20 ng/ml) for 5 min, and incubated with HPV16 PsVs for 5 min. Western blots show phosphorylated and total ERK1 and 2 for the indicated conditions; α-tubulin was used as a loading control. (**C**) The amount of phosphorylated ERK1 (black) and ERK2 (gray) is shown as a ratio of phosphorylated to total ERK form. Data for ERK1 (n = 8–9) compared to the corresponding ERK1 of the contr. si were analyzed with two sample t-test: p=3.85E-08 (ADAM17 si) and Welch two sample t-test: p=1.73E-05 (ADAM17 si + EGF), and the difference between ADAM17 si and ADAM17 si + EGF: p=3.15E-06. Data for ERK2 (n = 9)

*Figure 6 continued on next page*

*Figure 6 continued*

when compared to the corresponding ERK2 of control siRNA-treated cells (contr. si); Welch two sample t-test: p=1.61E-06 (ADAM17 si) and p=1.13E-03 (ADAM17 si + EGF), and the difference between ADAM17 si and ADAM17 si + EGF: p=2.62E-05. The values are given as mean ± SEM and the mean for contr. si was set to 100%. (D) Infection assay after EGF reconstitution. HaCaT cells were transfected with control (left panel) or ADAM17 siRNA (right panel) for 48 hr, left non-treated or treated with EGF (20 ng/ml) and infected with HPV16 PsVs. Samples shown on the left panel (n = 7) were analyzed using Wilcoxon rank sum test: p=2.33E-03 and on the right panel (n = 9) with two sample t-test: p=1.30E-06. Infection rate (in %) is given as mean ± SEM and the mean of EGF-non-treated control siRNA-treated cells was normalized to 100%. Because of simplicity, the data is shown in separate graphs (panels). (E) Shown is specific CD151-L1 PLA signal (red). HaCaT cells were depleted of ADAM17 and either left non-treated, or pre-treated with EGF (20 ng/ml) for 5 min and afterwards incubated with PsVs for 5 hr. (F) The ratio of PLA-positive signal (red) to nuclear signal (blue) (in %) is given as mean ± SEM. and the mean was normalized to EGF-non-treated control cells (shown in *Figure 2G*). Data (n = 128–152 images) were analyzed using Wilcoxon rank sum test: p=7.97E-06.

DOI: https://doi.org/10.7554/eLife.44345.014

The following figure supplement is available for figure 6:

**Figure supplement 1.** ADAM17 affects ERK1/2 in PsVs-non-exposed cells but PsVs have no effect on ERK1/2.

DOI: https://doi.org/10.7554/eLife.44345.015

*Horiuchi et al., 2007*). Both, AREG and TGF-α, were significantly reduced in the supernatant of ADAM17-depleted cells (*Figure 7A and B*), while HB-EGF was undetectable (not shown). Moreover, infection analyses showed that infection rates were significantly reduced when AREG or TGF-α neutralizing antibodies were added to the cells 1 hr before HPV16 PsVs addition (*Figure 7C*). These data verified the role of ADAM17 in the release of AREG and TGF-α and indicated a role of these growth factors in HPV16 PsVs infection.

Finally, we mimicked ADAM17 activity in ADAM17-depleted cells by adding the ADAM17 cleavage products HB-EGF, AREG, and TGF-α (*Figure 7D–G*). Supplying the medium with these GFs restored ERK phosphorylation. Moreover, GF treatment increased infection rates in the absence as well as in the presence of ADAM17 in a concentration-dependent manner (*Figure 7F and G*). The beneficial effect on infection rates was observed when the cells were grown under starvation conditions (in medium without FCS) in the presence (*Figure 7F*) and in the absence of ADAM17 (*Figure 7G*). Infection was not further supported by the addition of GFs if the cells were treated with control siRNA in full medium (containing FCS) (*Figure 7—figure supplement 1A*) but was supported in ADAM17-depleted cells (*Figure 7—figure supplement 1B*).

These findings show that ADAM17 sheds soluble biomolecules like HB-EGF, TGF-α, and AREG, which leads to activation of the ERK1/2 signaling pathway and successful HPV16 infection.

In conclusion, cells with functional ADAM17, or cells in the neighborhood of such cells, are susceptible for infection mediated by the cleavage of membrane-bound growth factor precursors. There is no specific growth factor. Instead ADAM17 acts by releasing a subset of different GFs. The GFs then activate the ERK signaling pathway, a precondition for the maturation of viral particle-carrying CD151/EGFR clusters that function as entry platforms.

## Discussion

In this study, we identify the ADAM17 protease as an important host cell factor in infections by oncogenic human papillomaviruses. ADAM17 mediates ectodomain shedding of membrane proteins, thereby releasing growth factors (GF). These released bioactive molecules bind to cell surface receptors of keratinocytes making them susceptible for infection. Our data suggest ADAM17-mediated release of GFs such as AREG, TGF-α or HB-EGF, known activators of ERK signaling as an important step in infection with oncogenic papillomaviruses. Activation of ERK signaling is a prerequisite for the formation of an HPV16-associated CD151/EGFR entry platform.

### The secondary entry receptor complex is a hallmark for infection

HPV infection follows a cascade of events including initial virus interaction with a primary receptor, followed by modifications of the viral capsid, and virus association to a secondary entry receptor complex. Only the secondary entry receptor complex enables for endocytic uptake into the infectious intracellular pathway (*Raff et al., 2013*; *Campos, 2017*). Up to now, the mechanisms underlying the formation of the secondary entry receptor complex are unknown. For instance, it is unclear

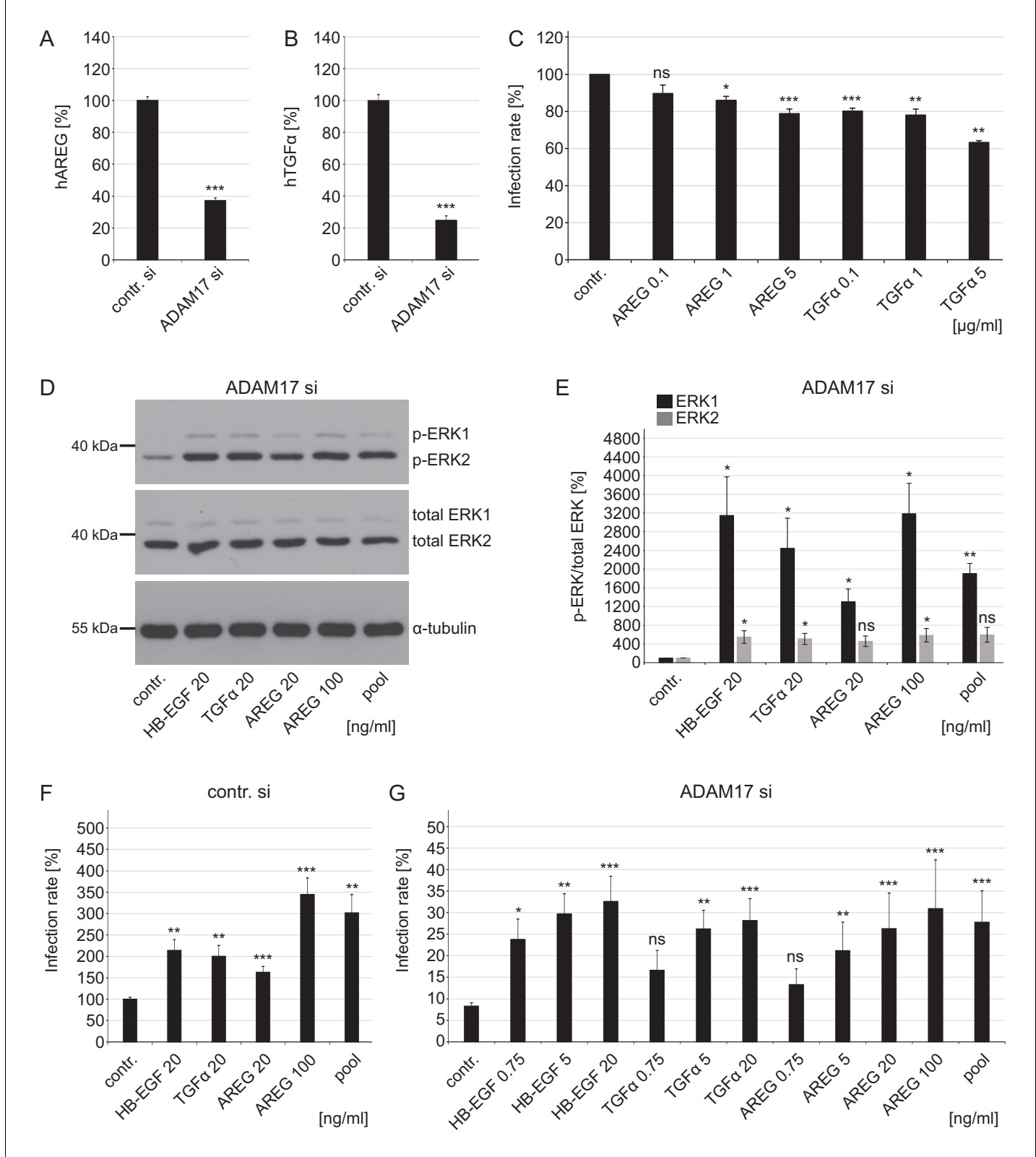

**Figure 7.** Growth factors affect ERK1/2 signaling and HPV16 infection rate in ADAM17-depleted cells. (**A**) ADAM17 depletion diminishes release of human AREG (ELISA). HaCaT cells were depleted of ADAM17 for 48 hr, afterwards starved in medium without FCS and 24 hr later supernatants were collected. Supernatants (n = 24) were analyzed with Wilcoxon rank sum test: p=3.06E-09. Release of AREG is given as mean ± SEM and the mean for control siRNA-treated cells (contr. si) was set to 100%. (**B**) Knockdown of ADAM17 decreases release of human TGFα (Ectodomain shedding assay).
*Figure 7 continued on next page*

*Figure 7 continued*

HeLa cells were treated with ADAM17-specific siRNA, 24 hr later transfected with alkaline phosphatase-tagged TGFα plasmid and the next day incubated in the fresh medium without FCS for 3 hr at 37˚C. Supernatants (n = 7) were analyzed with two sample t-test: p=1.83E-09. The release of transfected AP-tagged TGFα, used as readout for ADAM17 activity, is shown as a percentage and the mean for control siRNA-treated cells was set to 100%. (C) AREG and TGFα neutralizing antibodies significantly diminish HPV16 infection rate. HaCaT cells were pre-treated with specified antibodies murine IgG (contr.), AREG or TGFα for 1 hr, infected with PsVs and analyzed 24 hr later. Data for AREG (n = 6) were analyzed with Wilcoxon rank sum test: p=0.227 (AREG 0.1 ng/ml), Welch two sample t- test: p=0.011 (AREG 1 ng/ml) and two sample t-test: p=1.05E-04 (AREG 5 ng/ml). Data for TGFα (n = 6) were analyzed with two sample t-test: p=1.35E-06 (TGFα 0.1 ng/ml) and p=1.20E-03 (TGFα 1 ng/ml) and Wilcoxon rank sum test: p=4.77E-03 (TGFα 5 ng/ml). The values are given as mean ± SEM and the mean of all tested concentrations for murine IgG-treated cells (contr.) was set to 100%. (D) HaCaT cells were depleted of ADAM17 for 48 hr. Next, cells were starved for 4 hr in medium without FCS, either left non-treated or treated with depicted concentrations of different GFs for 15 min. Western blots show phosphorylated and total ERK1 and ERK2 for the indicated conditions; α-tubulin was used as a loading control. (E) The amount of phosphorylated ERK1 (in black) and ERK2 (in gray) is shown as a ratio of phosphorylated to total ERK form. Data for ERK1 (n = 4) were analyzed with Welch two sample t-test: p=0.034 (HB-EGF 20 ng/ml), p=0.036 (TGFα 20 ng/ml), p=0.022 (AREG 20 ng/ml), p=0.018 (AREG 100 ng/ml) and p=3.70E-03 (pool). Data for ERK2 (n = 4); Welch two sample t-test: p=0.047 (HB-EGF 20 ng/ml), p=0.042 (TGFα 20 ng/ml), p=0.051 (AREG 20 ng/ml), p=0.043 (AREG 100 ng/ml) and p=0.052 (pool). The values are given as mean ± SEM. and the mean for control-treated cells (contr.) was set to 100%. (F) Control siRNA-treated HaCaTs were starved for 2 hr in medium without FCS, treated with depicted concentration of different GFs for 15 min and infected with PsVs for 24 hr. Data (n = 7–10) were analyzed with Welch two sample t-test: p=0.13E-03 (HB-EGF 20 ng/ml), p=2.95E-03 (TGFα 20 ng/ml), p=1.00E-03 (AREG 20 ng/ml), 6.86E-04 (AREG 100 ng/ml) and p=3.03E-03 (pool). The values are given as mean ± SEM. and the mean for non-treated cells (contr.) was set to 100%. (G) ADAM17 siRNA-treated cells were treated as in (F). Data (n = 5–12) when compared to non-treated cells (contr.) were analyzed with Welch two sample t-test: p=0.014 (HB-EGF 0.75 ng/ml), p=3.59E-03 (HB-EGF 5 ng/ml), p=0.130 (TGFα 0.75 ng/m), p=4.09E-03 (TGFα 5 ng/ml), p=0.617 (AREG 0.75 ng/ml), p=3.23E-03 (AREG 5 ng/ml), p=3.23E-04 (AREG 20 ng/ml), p=1.08E-04 (AREG 100 ng/ml), p=1.08E-04 (pool), and with Wilcoxon rank sum test: p=1.48E-06 (HB-EGF 20 ng/ml), p=3.09E-06 (TGFα 20 ng/ml). The values are given as mean ± SEM. and the mean for non-treated cells (contr.) was set to 8.31%. Pool represents mixture of HB-EGF, TGFα and AREG used at final concentration of 20 ng/ml for each GF.

DOI: https://doi.org/10.7554/eLife.44345.016
The following figure supplement is available for figure 7:

**Figure supplement 1.** Growth factors increase HPV16 infection rate in ADAM17 siRNA-treated cells under full medium conditions.
DOI: https://doi.org/10.7554/eLife.44345.017

what triggers the assembly of the secondary entry receptor complex, that contains CD151 and EGFR as functional components (*Mikuličić and Florin, 2019*).

Examining at nanoscale resolution the secondary entry receptor complex component CD151, briefly after binding, we find viral particles associated with CD151 clusters and over the next hours an enrichment of CD151 at the virus binding site. Because viral particles are almost immobile when attached to the cell body, as intensively investigated and discussed before (*Schelhaas et al., 2008*), we suggest that CD151 is recruited to the virus binding site. Only in the presence of ADAM17, PsVs trigger the internalization of preferentially larger clusters. Besides, the proximity ligation assay shows an ADAM17-dependent close association between viral particles and CD151.

In contrast, ADAM17 neither affects virus binding to the cell surface, nor influences the release of L1 protein. It is possible that a large fraction of viral particles is not released from the cell surface, as expected from the primary receptor/secondary receptor model, but can remain at or in close proximity to the initial binding site.

Together, these findings suggest that after PsVs binding to the cell body, it takes several hours for the recruitment of CD151 clusters and maturation of a functional virus entry platform. Apart from CD151, it is likely that other components enrich at the viral binding site as well, possibly GF/GFR-complexes, integrins, annexins and actin for endocytosis.

## ADAM17-released growth factors

In the co-culture transwell assay, modified viral particles are transferred only from the donor cells to the recipient cells. In contrast, bioactive molecules, like growth factors, exchange in both directions.

The co-culture assay provides a couple of important observations. First, ADAM17 acts after virus primary binding/release, because infection is almost blocked in ADAM17-depleted recipient cells, receiving viral particles from control donor cells. The second observation is that ADAM17-depleted donor cells infect at a lesser extent their respective recipient cells, and in parallel, that control donor cells are less infected when recipient cells are ADAM17-depleted. This mutual influence is explained by an exchange of bioactive molecules between donor and recipient cells in both directions. The

association of bioactive molecules with the viral capsid as proposed by the Ozbun group (*Surviladze et al., 2012*) may increase the effect.

Consistently, additional findings pointed toward an involvement of the EGFR as the receptor that mediates the action of these biomolecules. For example, EGF increases the infection rate of ADAM17-depleted cells, EGF increases the proximity between L1 and CD151 after ADAM17 knockdown, and ADAM17 knockdown diminishes the overlap between L1 and EGFR. However, EGF is not classically generated by the action of ADAM17 (*Sahin et al., 2004*; *Horiuchi et al., 2007*). Therefore, we tested the involvement of additional EGFR ligands generated by ADAM17 ectodomain shedding. We find less AREG and TGF-α in the supernatant of ADAM17-depleted cells, corroborated by a reduction of the infection rate after treatment with AREG and TGF-α neutralizing antibodies. Adding ADAM17 cleavage products such as AREG, TGF-α, and HB-EGF to the supernatant of serum-starved or ADAM17-depleted cells increases ERK1/2 phosphorylation and infection. The effect of EGFR, AREG or TGF-α inhibition is much lower than that of the ADAM17 siRNA, therefore, we have to assume that ADAM17 activates not only the EGFR but also the additional GFRs.

Together, these results provide a link between ADAM17-mediated GF-shedding, activation of the ERK1/2 signaling pathway and HPV16 infection. This hypothesis combines previously reported observations that ADAM17 activity regulates multiple cell signaling pathways by the release of bioactive proteins like EGFR ligands (*Gooz, 2010*; *Sahin et al., 2004*; *Göoz et al., 2006*) and cell signaling plays a role in early events of HPV16 entry (*Schelhaas et al., 2012*; *Surviladze et al., 2012*; *Surviladze et al., 2013*; *Abban and Meneses, 2010*). In line with our results, it has been shown by the Ozbun group that HPV16 particles associate with GFs like AREG or HB-EGF forming an infectious complex able to bind to the HPV16 entry receptor (*Surviladze et al., 2012*; *Surviladze et al., 2015*). Due to the plethora of ADAM17 substrates (*Gooz, 2010*; *Reiss and Saftig, 2009*; *Edwards et al., 2008*) and the complex crosstalk between signaling pathways, also, ADAM17-dependent shedding of additional membrane-bound precursors and cleavage of integral membrane proteins might be involved in this scenario.

## Internalization of the HPV/CD151 complex and L1 cleavage

CD151 is co-internalized with viral particles during virus endocytosis (*Scheffer et al., 2013*), a mechanism supposed to decrease the content of CD151 in the plasma membrane. Accordingly, we observed a reduction of CD151 levels in the plasma membrane only in the presence of PsVs and ADAM17. Furthermore, we observed an ADAM17-dependent cleavage of the L1 protein. It has been described that L1 cleavage occurs by the action of specific proteases on the cell surface and after internalization via the infectious entry pathway (*Cerqueira et al., 2015*). However, our findings suggest that ADAM17 has no direct effect on extra- or intracellular processing of the virus particle or virus internalization. Instead, affected L1 cleavage is a consequence of virus internalization into a non-infectious pathway and reduced capsid interaction with proteases resident within the entry platform complex or in the endosomal pathway. Thus, ADAM17 mediates virus association with the functional HPV-endocytic platform enabling host cell entry via the infectious pathway.

## A generalized scheme

Consistent with the idea of an entry platform containing EGFR (*Surviladze et al., 2012*), we showed a significant reduction of the HPV16-EGFR association upon ADAM17 depletion. In agreement with this, it has been recently shown that EGFR acts as a cofactor for HCV entry by promoting specific tetraspanin interactions with other proteins that result in the assembly of the HCV entry receptor complex (*Zona et al., 2013*; *Bruening et al., 2018*). This points toward a general scheme in viral uptake independent from the virus structure, further supported by the notion that tetraspanins stand for essential components of entry receptor complexes for corona-, influenza-, hepatitis C and other viruses (*Zona et al., 2013*; *Earnest et al., 2015*; *Fast et al., 2017*; *Florin and Lang, 2018*). We propose a model in which GF/GFR complex formation and GFR-mediated ERK signaling is a prerequisite for the formation of virus-associated tetraspanin entry platforms for many types of viruses.

It seems plausible that the GFR-dimerization triggered by the interaction of growth factors with its cognate receptors induces local accumulation of the associated tetraspanins, including CD151. Thus, clustering of cell factors might explain a more than two-fold increase in the CD151

microdomain size and internalization of the virus-associated entry platform. Furthermore, tetraspanin CD151 regulates integrin activities by influencing their positioning within tetraspanin-enriched microdomains (*Berditchevski et al., 2001*; *Stipp, 2010*; *Yang et al., 2004*). Crowding of CD151 at the sites where virus particles are bound results in co-accumulation of additional virus interacting molecules, laminin and integrin complexes, which is in line with our previous analysis on CD151 mutants demonstrating the importance of CD151/integrin complex formation in HPV16 infection (*Scheffer et al., 2013*). Furthermore, the membrane-associated protein annexin A2 has been identified in virus/GFR complexes and is utilized by many viruses during cell entry, including HPV16 (*Mikuličić and Florin, 2019*; *Dziduszko and Ozbun, 2013*; *Taylor et al., 2018*).

In summary, much work has been done to unveil the process of human papillomavirus entry. Although for this puzzling process diverse host cell factors and many steps have been proposed so far, the precise description still remains controversial. In order to deepen our understanding of these early steps of infection, we investigated the requirements for HPV16 entry platform formation. We propose a model in which ADAM17 acts by the shedding of growth factors. These molecules trigger ERK signaling activation and thereby mediate the co-clustering of preformed larger CD151-EGFR microdomains and HPV16 in the plasma membrane. These events lead to the assembly of a functional virus entry platform, priming of the HPV capsid proteins, endocytosis into the infectious pathway and eventually infection.

# Materials and methods

## Key resources table

| Reagent type (species) or resource | Designation | Source or reference | Identifiers | Additional information |
|---|---|---|---|---|
| Cell line (*Homo sapiens*) | HaCaT cells | Cell Lines Services (CLS) | Cat. #: 300493; RRID:CVCL_0038 | human immortalized keratinocytes (from adult skin) |
| Cell line (*Homo sapiens*) | HeLa cells | German Resource Center of Biological Material (DSMZ) | Cat. #: ACC 57; RRID:CVCL_0030 | human cervical carcinoma cell line |
| Cell line (*Homo sapiens*) | NHEK cells | PromoCell | Cat. #: C-12002 | normal human epidermal keratinocytes |
| Antibody | anti-ADAM10 (rabbit polyclonal) | Merck Millipore | Cat. #: AB19026; RRID:AB_2242320 | WB (1:750) |
| Antibody | anti-ADAM17 (rabbit polyclonal) | Merck Millipore | Cat. #: AB19027; RRID:AB_91097 | WB (1:1000) |
| Antibody | anti-AREG (goat polyclonal) | R and D Systems | Cat. #: AF262; RRID:AB_2243124 | |
| Antibody | anti-CD151 (mouse monoclonal) | Bio-Rad | Bio-Rad: MCA1856; RRID:AB_2228964 | IHC (1:100) |
| Antibody | anti-EGFR (rabbit monoclonal) | Cell Signaling | Cat. #: 4267; RRID:AB_2246311 | IHC (1:100) |
| Antibody | anti-EGFR; Cetuximab | Merck | 3023710001 | |
| Antibody | anti-L1 K75 (rabbit polyclonall) | PMID: 15543569 | | WB (1:10000); IHC (1:1000), flow cytometry (1:500) |
| Antibody | anti-L1 16L1-312F (mouse monoclonal) | PMID: 17640876 | | WB (1:350); IHC (1:10) |
| Antibody | anti-L1 33L1-7 (mouse monoclonal) | PMID: 7996132 | | WB (1:350) |
| Antibody | anti-mouse Alexa Fluor 488 (goat polyclonal) | Molecular Probes (Invitrogen) | Cat. #: A-11029; RRID:AB_138404 | IHC (1:450) |

*Continued on next page*

*Continued*

| Reagent type (species) or resource | Designation | Source or reference | Identifiers | Additional information |
|---|---|---|---|---|
| Antibody | anti-mouse Alexa Fluor 546 (goat polyclonal) | Molecular Probes (Invitrogen) | Cat. #: A-11030; RRID:AB_144695 | IHC (1:450) |
| Antibody | anti-mouse Alexa Fluor 647 (donkey polyclonal) | Molecular Probes (Invitrogen) | Cat. #: A-31571; RRID:AB_162542 | IHC (1:200) |
| Antibody | anti-p44/42 MAPK (rabbit monoclonal) | Cell Signaling | Cat. #: 4695; RRID:AB_390779 | WB (1:2000) |
| Antibody | anti-p44/42 MAPK-Thr202, Tyr204 (rabbit monoclonal) | Cell Signaling | Cat. #: 4370; RRID:AB_2315112 | WB (1:2000) |
| Antibody | anti-rabbit Alexa Fluor 488 (goat polyclonal) | Molecular Probes (Invitrogen) | Cat. #: A-11034; RRID:AB_2576217 | IHC (1:450) |
| Antibody | anti-rabbit Alexa Fluor 546 (goat polyclonal) | Molecular Probes (Invitrogen) | Cat. #: A-11035; RRID:AB_143051 | IHC (1:450) |
| Antibody | anti-rabbit Alexa Fluor 594 (donkey polyclonal) | Molecular Probes (Invitrogen) | Cat. #: A-21207; RRID:AB_141637 | IHC (1:200) |
| Antibody | anti-TGFα (goat polyclonal) | R and D Systems | Cat. #: AF-239; RRID:AB_2201779 | |
| Antibody | anti-α-tubulin (mouse monoclonal) | Sigma-Aldrich | Cat. #: T5168; RRID:AB_477579 | WB (1:10000) |
| Antibody | anti-β-actin (mouse monoclonal) | Sigma-Aldrich | Cat. #: A5441; RRID:AB_476744 | WB (1:10000) |
| Antibody | control IgG (mouse) | Sigma-Aldrich | Cat. #: I5381; RRID:AB_1163670 | |
| Antibody | HRP anti-rabbit (polyclonal) | Jackson ImmunoResearch | Cat. #: 111-035-003; RRID:AB_231356 | WB (1:10000) |
| Antibody | HRP anti-mouse (polyclonal) | Jackson ImmunoResearch | Cat. #: 115-035-003; RRID:AB_10015289 | WB (1:10000) |
| Recombinant DNA reagent | AP-TGFα | PMID: 17079736 | | provided by Dr. Carl P. Blobel (Hospital for Special Surgery, New York, USA) |
| Recombinant DNA reagent | CD151-CFP | PMID: 23302890 | | |
| Recombinant DNA reagent | CD151-GFP | PMID: 23302890 | | provided by Dr. Dr. Xin A. Zhang (Oklahoma City, USA) |
| Sequenced-based reagent | siRNA (in this paper ADAM10 #1) | Sigma-Aldrich | | sequence GGACAAACU UAACAACAAU |
| Sequenced-based reagent | siRNA (in this paper ADAM10 #2) | Sigma-Aldrich | | sequence UACACCAGUCAU CUGGUAUUUCCUC |
| Sequenced-based reagent | siRNA (in this paper ADAM17 #1) | Invitrogen | | sequence GGAAGCUGACCUG GUUACAACUCAU |
| Sequenced-based reagent | siRNA (in this paper ADAM17 #2) | Invitrogen | | sequence CCAGGGAGGGAA AUAUGUCAUGUAU |

*Continued on next page*

*Continued*

| Reagent type (species) or resource | Designation | Source or reference | Identifiers | Additional information |
|---|---|---|---|---|
| Peptide, recombinant protein | EGF-Alexa Fluor 488 complex | Thermo Fischer Scientific | Cat. #: E13345 | |
| Peptide, recombinant protein | human ADAM17 | R and D Systems | Cat. #: 930-ADB | |
| Peptide, recombinant protein | human AREG | Peprotech | Cat. #: 100-55B | |
| Peptide, recombinant protein | human HB-EGF | Roche | Cat. #: 259-HE | |
| Peptide, recombinant protein | human TGFα | Biolegend | Cat. #: 589904 | |
| Commercial assay, kit | CytoTox-ONE Homogeneous Membrane Integrity Assay | Promega | Cat. #: G7890 | |
| Commercial assay, kit | DuolinkIn Situ Orange Starter Kit Mouse/Rabbit | Sigma-Aldrich | Cat. #: DUO92102 | |
| Commercial assay, kit | Human Amphiregulin DuoSet ELISA kit | R and D Systems | Cat. #: DY262 | |
| Chemical compound | GI254023X | Tocris Bioscience | Cat. #: 3995 | |
| Chemical compound | GW280264X | Aobious | Cat. #: AOB3632 | |
| Chemical compound | TAPI-0 | Sigma-Aldrich | Cat. #: SML1292 | |
| Other | luciferase assay buffer | This paper | | see Materials and methods |
| Software, algorithm | ImageJ | ImageJ (http://imagej.nih.gov/ij/) | | |
| Software, algorithm | Statistical Software R (2017, version 3.3.3) | R (http://www.R-project.org/) | | R: A language and environment for statistical computing. R Foundation for Statistical Computing, Vienna, Austria |

## Antibodies and plasmids

HPV16 L1 mouse monoclonal antibodies (mAb) 16L1-312F and 33L1-7, as well as rabbit polyclonal antibody (pAb) K75 have been previously described (*Knappe et al., 2007*; *Rommel et al., 2005*; *Sapp et al., 1994*). ADAM17-specific (AB19027) and ADAM10-specific (AB19026) rabbit pAb was purchased from Merck Millipore (Darmstadt, Germany). β-actin (A5441) and α-tubulin (B-5-1-2)-specific mouse mAbs were from Sigma-Aldrich (St. Louis, MO). CD151-specific mouse mAb (11G5a) was obtained from Bio-Rad (Munich, Germany). Rabbit monoclonal antibodies specific for EGFR (D38B1), total ERK1/2 (p44/42 MAPK; clone 137F5) and phosphorylated ERK (Phospho-p44/42 MAPK; clone D13.14.4E) were obtained from Cell Signaling (Leiden, Netherlands). Horseradish peroxidase-coupled (HRP and POD) secondary antibodies for immunoblot were purchased from Jackson ImmunoResearch Europe Ltd. (Cambridgeshire, UK). Secondary antibodies for immunofluorescence detection on Zeiss Axiovert 200 M microscope (goat-derived Alexa Fluor 488, 546 and 647 antibodies) and for STED analyses [donkey anti-rabbit Alexa Fluor 594 (cat# A21207) and donkey anti-mouse Alexa Fluor 647 (cat# A31571)] were from Molecular Probes (Invitrogen, Carlsbad, CA). Neutralization antibody for EGFR (Cetuximab) was purchased from Merck, control mouse IgG antibody (cat# I5381) was from Sigma-

Aldrich, human amphiregulin (cat# AF262) and transforming growth factor α (cat# AF-239) were purchased from R and D Systems (Minneapolis, MN, Canada). Cyan fluorescent protein (CFP)-tagged CD151 (CFP-CD151) was generated by isolating the CD151 gene from pEGFP-C1/CD151 with BspEI and XmaI and inserted into pECFP-C1 (Clontech) (*Scheffer et al., 2013*). AP (alkaline phosphatase)-tagged TGFα [(AP-TGFα) was provided by Dr Carl P. Blobel (*Horiuchi et al., 2007*).

## Production of pseudoviruses

HPV16, −18,–31 pseudoviruses (PsVs) were prepared as previously described (*Buck et al., 2004*). In short, expression plasmids carrying codon-optimized L1 and L2 expression vectors were co-transfected with a pcDNA 3.1 (+)-Luciferase reporter plasmid (*Schneider et al., 2011*) into HEK 293TT cells (*Bund et al., 2014*; *Spoden et al., 2012*). Mock preparation was prepared in parallel to PsVs preparation except that HEK 293TT cells were transfected with Luciferase reporter plasmid only. HEK 293TT (human embryonic kidney cell line) cells were provided by Chris Buck (Bethesda, MD) (*Buck et al., 2004*; *Buck et al., 2005*). 48 hr after transfection, cells were lysed and the pseudoviruses were purified from the cell lysates using OptiPrep (Sigma-Aldrich) gradient centrifugation. Quantification of pcDNA3.1 (+)-Luciferase positive PsVs was performed as described (*Scheffer et al., 2013*; *Bund et al., 2014*).

For HPV16 PsVs labeling, HPV16 PsVs were incubated with a 10-fold molar excess of Alexa Fluor 488 succinimidylester over that of the major capsid protein L1 for 1 hr at room temperature, as described previously (*Scheffer et al., 2013*; *Schelhaas et al., 2008*). PsVs were separated from the labeling reagent by size-exclusion chromatography using NAP5 columns (GE Healthcare, Chicago, IL).

## Cells

The human cervical carcinoma cell line (HeLa) was obtained from the German Resource Center of Biological Material [(DSMZ), Braunschweig, Germany]. Human immortalized keratinocytes (HaCaT) were purchased from Cell Lines Services [(CLS), Eppelheim, Germany]. The cells were grown at 37°C in Dulbecco's modified Eagle's medium (DMEM, Invitrogen) supplemented with 1% Glutamax (Invitrogen), 10% fetal calf serum (FCS, Biochrom AG, Berlin, Germany), 1% Eagle's minimum essential medium (MEM) non-essential amino acids (GE Healthcare) and antibiotics (Fresenius Kabi, Bad Homburg vor der Hoehe, Germany). For the experiments, cells were grown in the absence of antibiotics in DMEM supplemented with 1% Glutamax and either with or without 10% FCS. Cell lines were authenticated using Short Tandem Repeat (STR) analysis (Microsynth, Lindau, Germany). Cell lines were tested negative for mycoplasma (MycoAlert PLUS Mycoplasma Detection Kit, Lonza, Koeln, Germany) and by Microsynth Real-Time PCR analysis (Microsynth, Lindau, Germany). Normal Human Epidermal Keratinocytes (NHEK) were purchased from PromoCell (Heidelberg, Germany) and cultivated according to the manufacturer's instructions.

## siRNA-mediated knockdown

The following ADAM17-specific siRNAs were obtained from Invitrogen: ADAM17#1 (GGAAGC UGACCUGGUUACAACUCAU), ADAM17#2 (CCAGGGAGGGAAAUAUGUCAUGUAU) and ADAM17#pool as a mixture and with the equal amounts of ADAM17#1 and ADAM17#2 siRNAs. All-Stars Negative Control siRNA was used as non-silencing control and was obtained from Qiagen (Hilden, Germany). The following ADAM10-specific siRNAs were obtained from Sigma-Aldrich: ADAM10#1 (GGACAAACUUAACAACAAU) and ADAM10#2 (UACACCAGUCAUCUGGUAUUUCC UC). Cells were transfected with 30 nM siRNA for 48 hr using Lipofectamine RNAiMAX (Invitrogen) according to the manufacturer's instructions.

## Infection assays

Pseudovirus infection assays were performed as described previously (*Spoden et al., 2012*). Briefly, HeLa, HaCaT or NHEK cells were transfected with control or ADAM-specific siRNA for 48 hr. Then, the cells were exposed to ≈100 (or ≈500 for NHEK) viral genome equivalents (vge) of PsVs. 24 hr post infection cells were lysed and luciferase activity (using Luciferase substrate buffer: 1 mM coenzyme A, 50 mM luciferin, 50 mM ATP, 0.5 M EDTA, 1 M DTT, 0.5 M Tris-HCl, pH 7.8, 1 M MgSO₄)

as gene transduction efficiency was measured and normalized to lactate dehydrogenase (LDH) measurements (CytoTox-ONE Homogeneous Membrane Integrity Assay, Promega, Fitchburg, MA).

Both, luciferase and LDH activities were measured by the Tristar LB 941 luminometer (Berthold Technologies, Bad Wildbad, Germany). For inhibition studies, HaCaT cells were pre-treated with specified inhibitor for 1 hr before HPV16 PsVs addition. We used TAPI-0 as ADAM17 metalloproteinase, collagenase and gelatinase inhibitor [(10 µM), Sigma-Aldrich], GI254023X (GI) as potent ADAM10 inhibitor (*Sommer et al., 2016*; *Ludwig et al., 2005*) [(3 µM), Tocris Bioscience, Bristol, UK] or the hydroxamate-based GW280264X (GW) inhibitor as mixed ADAM17/ADAM10 suppressor (*Sommer et al., 2016*; *Ludwig et al., 2005*) [(3 µM), Aobious, Gloucester, MA]. The infection rate was assessed 24 hr later. Control cells were treated with dimethyl sulfoxide (Carl Roth, Karlsruhe, Germany) only. For the infection assay using epidermal growth factor [EGF-Alexa Fluor 488 complex (Thermo Fischer Scientific, Waltham, MA)] we pre-treated the cells with EGF (20 ng/ml) for 1 hr at 37°C prior addition of the virus. For neutralization experiments with human EGFR, AREG and TGFα, cells were incubated in medium with FCS and pre-treated with specific neutralizing antibody for 1 hr at 37°C followed by infection with PsVs for additional 24 hr. For recovery studies with growth factors HaCaT cells were either starved in medium without FCS for 2 hr, pre-treated with human HB-EGF (cat# 259-HE, Roche), TGFα (cat# 589904, Biolegend, Koblenz, Germany) or AREG (cat# 100-55B, Peprotech, Hamburg, Deutschland) for 15 min at 37°C and infected with PsVs for another 24 hr, or cultured in medium with FCS, treated with specified growth factor for 15 min and followed by infection with PsVs.

## Control infection assay

HeLa and HaCaT cells were transfected either with control, luciferase or ADAM17 siRNA. One day later, cells were transfected with luciferase expressing plasmid (pcDNA3.1-luciferase from Invitrogen) and 24 hr later analyzed for luciferase signal. Luciferase siRNA with the following sequence CTTACGCTGAGTACTTCGAdT was used as a positive control (Sigma-Aldrich).

## Cell binding assay

HaCaT cells were transfected with control or ADAM17 siRNAs for 48 hr. To analyze virus-cell-binding efficiency, cells were subsequently incubated with 100–500 vge HPV16 PsVs for 1 hr at 4°C, extensively washed with PBS to remove unbound virus and detached with 0.05% trypsin/2.5 mM EDTA. Surface-bound particles were stained with anti-L1 polyclonal antibody K75 in 0.5% FCS/PBS for 30 min at 4°C followed by staining with secondary antibody anti-rabbit Alexa Fluor 488 in 0.5% FCS/PBS for 20 min at 4°C. The amount of surface particles was validated using FACScan flow cytometer and CellQuest3.3 software (Becton Dickinson, East Rutherford, NJ, USA) as described before (*Scheffer et al., 2013*; *Wüstenhagen et al., 2016*).

## L1 release in the supernatant

HaCaT cells were transfected either with control or with ADAM17 siRNAs (ADAM17#pool). After 48 hr, cells were incubated with 500–1000 HPV16 vge for 15 min at 4°C. Next, the cells were washed with ice-cold FCS and incubated in fresh medium for 4 hr at 37°C. Afterwards, the supernatant was transferred into siliconized tubes, samples were centrifuged, transferred into fresh tubes and proteins were precipitated overnight at −20°C using acetone. The next day, samples were lysed in SDS sample buffer and analyzed by western blot.

## Western blot analysis

For detection of the major capsid viral protein L1, HaCaT cells were washed with phosphate-buffered saline (PBS), lysed in sodium dodecyl sulfate (SDS) sample buffer (250 mM Tris-HCl, 0.3% glycerine, 0.1% SDS and 10% 2-mercaptoethanol) and denatured at 95°C. The samples were electrotransferred onto nitrocellulose membrane (GE Healthcare) and blocked with 5% milk powder in PBS. Afterwards, the membrane was incubated with primary antibody overnight at 4°C, next day washed in PBST (Phosphate-buffered saline containing 0.1% Tween-20) and stained with horseradish peroxidase (HRP)-conjugated secondary antibody. Detection was carried out using the Western Lightning Plus ECL detection reagent (PerkinElmer, Waltham, MA) and the signals were recorded on scientific imaging Super RX-N films (Fujifilm, Tokio, Japan). For ADAM17 and ERK

proteins, cells were lysed in lysis buffer containing 5 mM Tris-HCl pH 7.4, 1 mM EGTA, 250 mM sucrose and 1% Triton X-100. For ADAM17 analyses, the lysis buffer was supplemented with cOmplete protease inhibitor cocktail (Roche, Penzberg, Germany) and 10 mM 1,10-phenanthroline monohydrate to prevent ADAM autocleavage (Schlöndorff et al., 2000), and for ERK studies additionally with phosphatase inhibitor cocktail PhosSTOP (Roche). The cells were lysed applying three freeze-thaw cycles (freezing at −80°C and thawing on 4°C) and denatured at 95°C for 5 min in SDS sample buffer. Equal amounts of protein were loaded on SDS–PAGE gel. The samples were electrotransferred either onto polyvinylidene difluoride [(Hybond-P), GE Healthcare] or nitrocellulose membrane and blocked with 5% milk powder in Tris-buffered saline (TBS). After incubation with primary antibodies proteins were detected using either POD- or HRP-conjugated secondary antibody. Detection was carried out using Amersham ECL detection system (GE Healthcare) or Western Lightning Plus ECL detection reagent (PerkinElmer). Signals were recorded either by a luminescent image analyzer Fusion FX7 imaging system (PEQLAB Biotechnologie, Erlangen, Germany) or scientific imaging X-ray films for western Blot detection Super RX-N (Fujifilm, Duesseldorf, Germany).

## Proteolytic processing of L1

HaCaT cells were transfected with control siRNA or ADAM17 siRNA pool for 48 hr. Afterwards, cells were incubated with 500–1000 HPV16 vge for 1 hr at 4°C, washed with medium supplemented with 10% FCS and incubated for another 24 hr. Subsequently, cells were washed with PBS and lysed in sodium dodecyl sulfate (SDS) sample buffer in denaturing conditions. In the experiment with recombinant human ADAM17 (rhADAM17) protein (cat# 930-ADB, R and D Systems), we used HaCaTs incubated with 500–1000 HPV16 vge as a positive control for L1-specific proteolytic products. In parallel, we prepared a mixture of HPV16 PsVs and rhADAM17 in the assay buffer recommended for optimal protein activity (25 mM Tris, 2.5 µM ZnCl2, 0.005% Brij-35, pH 9.0) and following manufacturer's recommendations. 24 hr later, the cells and the PsVs-rhADAM17 mixtures were directly lysed in SDS sample buffer and L1 proteolytic processing was analyzed by western blot.

## Proteinase K protection assay

Proteinase K protection assay was performed as described previously (Wüstenhagen et al., 2016; Milne et al., 2005). In brief, HaCaT cells were transfected with control siRNA or a pool of two different ADAM17-specific siRNAs for 48 hr. The cells were infected with 500–1000 HPV16 vge for 1 hr, washed two times with medium supplemented with 10% FCS and without antibiotics and incubated for another 24 hr. Afterwards, HaCaTs were washed two times with PBS and incubated with DMEM without FCS and antibiotics but supplemented with 20 µg/ml of Proteinase K (Sigma-Aldrich) for 15 min at 37°C in order to degrade viral particles still present on the cell surface. Control cells were treated with cell culture medium alone. Digestion was stopped by addition of 2 mM phenylmethylsulfonyl fluoride (PMSF). Subsequently, cells were washed with PBS, lyzed in SDS sample buffer and processed for western blot. Cytohalasin D (cytoD) was used as a control for blocked virus entry. CytoD prevents endocytosis by inhibiting actin polymerization.

## Enzyme-linked immunosorbent assay (ELISA)

To test effect of ADAM17 depletion on release of human amphiregulin (AREG) HaCaT cell were transfected with control or ADAM17-specific siRNA. Two days later, the medium was exchanged to medium without FCS and 24 hr later the supernatants were collected, centrifuged on 15,000 g for 5 min at 4°C, supplemented with cOmplete protease inhibitor cocktail and processed for analysis. Samples were tested with Human Amphiregulin DuoSet ELISA kit (DY262) from R and D Systems according to the manufacturer's instructions. As substrate solution BM Blue POD Substrate from Roche was applied. The measurement was performed at 450 nm and the correction wavelength at 540 nm.

## Ectodomain shedding assay

Ectodomain shedding assay (TGFα shedding assay) is in detail described previously by Inoue et al. (Inoue et al., 2012). In brief, HeLa cells were transfected with ADAM17 siRNA (48 hr before the assay) in order to decrease intrinsic ADAM17 levels. One day after siRNA transfection, HeLa cells were transfected with AP-TGFα plasmid using polyethylenimine [(PEI), Sigma-Aldrich]. 24 hr

after, the medium was replaced by fresh DMEM without FCS and cells were either exposed to the rhADAM17 protein or left non-treated. After incubation of 3 hr at 37°C, supernatants were collected and the cells were lysed in lysis buffer containing 10 mM 1,10-phenanthroline, 1 mM EDTA and 2.5% Triton X-100 in water. The phenanthroline has been shown as a metalloprotease inhibitor of ADAM17 catalysis .(*Schlöndorff et al., 2000*). For experiment with rhADAM17, the changes in the AP activity correlate with the activity of rhADAM17 that is responsible for shedding of the membrane-bound TGFα. The AP activity was assessed after administration of AP substrate 4-nitrophenyl phosphate (Sigma-Aldrich) by measuring absorbance at 405 nm. The readout was performed on Multiskan RC V1.5–0 (Labsystems, Helsinki, Finland) and using GENESIS software.

## Immunofluorescence microscopy (for analyses on Zeiss Axiovert microscope)

Cells were incubated with 100–500 HPV16 PsVs particles per cell, washed in PBS and fixed with 2% paraformaldehyde (PFA) prepared in PBS for 10 min at room temperature (RT). For EGFR co-staining, the cells were fixed with 2% PFA and treated with 0.2% Triton X-100 in PBS for 2 min at RT. Fixed cells were stained using primary, Alexa-conjugated secondary antibodies, and Hoechst33342 (Invitrogen). Fluorescence imaging was performed using a Zeiss Axiovert 200 M microscope equipped with a Plan-Apochromat 100x (1.4 NA) and Axiovision deconvolution and colocalization software 4.7 (Carl Zeiss, Jena, Germany). For determining amount of L1 (viruses) colocalizing either with CD151 or EGFR at 5 hr after PsVs exposure, we used randomly chosen rectangular areas of ~130 $\mu m^2$ and normalized assessed L1 pixels colocalizing with CD151 to the total L1 pixels.

## Duolink proximity ligation assay (PLA)

HaCaT cells were transfected with specific siRNA for 48 hr and either left non-treated or treated with soluble EGF (20 ng/ml) for 5 min before incubation with 100–500 HPV16 PsVs particles per cell. The next day, cells were fixed with 2% PFA, washed in PBS and processed for PLA (*Söderberg et al., 2006*). We used Duolink In Situ Orange Starter Kit Mouse/Rabbit (Sigma-Aldrich). Briefly, fixed cells were incubated with Duolink Blocking Solution for 1 hr at 37°C, followed by staining with primary antibodies recognizing CD151 and L1 of HPV16 dissolved in Duolink Antibody Diluent for another 1 hr at 37°C. Next, samples were washed in Wash buffer A that was prepared according to the manufacturer's recipe. The cells were incubated with the secondary antibodies conjugated with the DNA probes in Dulink Antibody Diluent solution for another 1 hr at 37°C. Then, samples were incubated with Ligation solution for 30 min at 37°C. Subsequently, samples incubated with the Amplification solution containing DNA polymerase for the rolling circle amplification (RCA) for 100 min at 37°C. Finally, the samples were washed and mounted with Duolink In Situ Mounting Media with Dapi. The signals ($\lambda_{em}$576 nm) were analyzed on Zeiss Axiovert 200 M microscope equipped with a Plan-Apochromat 100x (1.4 NA). For determining PLA signals, we normalized assessed L1 pixels to the pixels recorded for DAPI.

## Immunostaining for STED microscopy

HaCaT cells were cultured on poly-L-lysine coated glass coverslips. Cells were washed followed by fixation for 30 min at RT with 4% PFA in PBS (*Figure 3*), or incubated with 100–500 vge PsVs for 5 hr, washed, membrane sheets were generated and fixed (*Figure 4*). For *Figure 2* cells were incubated for 15 min with PsVs, washed and fixed, or further incubated up to 4:45 hr and then washed and fixed. As indicated, we used cells transfected as described above with control siRNA or a mixture of the two ADAM17-specific siRNAs. Plasma membrane sheets were generated applying 100 ms ultrasound sonication pulses essentially as described previously (*Homsi et al., 2014*), with the difference that several pulses at different coverslip locations were applied. After fixation samples were quenched with 50 mM $NH_4Cl$ in PBS for 30 min. Samples were permeabilized with 0.1% Triton X-100 in PBS for 2 min for cells and 45 s for membrane sheets to provide the access to epitopes present between the membrane outer leaflet and the glass coverslip. Afterwards, the samples were blocked with 3% BSA in PBS for 30 min. Samples were incubated for 1 hr at 37°C (whole cells) or overnight at 4°C (membrane sheets) with primary antibodies diluted in 3% BSA in PBS. Employed primary antibodies included mouse monoclonal 11G5a (see above) for CD151, rabbit polyclonal K75 (see above) for L1, and rabbit monoclonal D38B1 (see above) for EGFR. Afterwards, samples were washed with

PBS and incubated at RT with secondary antibodies donkey-anti-rabbit Alexa Fluor 594 and donkey-anti-mouse Alexa 647 in 3% BSA in PBS for 1 hr. Finally, samples were washed and mounted on microscopy slides employing ProLong Gold antifade
mountant medium (Invitrogen).

## STED microscopy

Two-color STED micrographs were acquired using a 4-channel easy3D superresolution STED optics module (Abberior Instruments, Goettingen, Germany) coupled with an Olympus IX83 confocal microscope (Olympus, Tokyo, Japan) and equipped with an UPlanSApo 100x (1.4 NA) objective (Olympus, Tokyo, Japan) (available in the LIMES imaging facility). Two-color STED-microscopy was realized by sequential imaging of the two channels, using pulsed 561 nm and 640 nm lasers for the excitation of Alexa 594 and 647, respectively. For depletion, a pulsed 775 nm STED laser was used. Signals emitted from Alexa Fluor 594 and Alexa 647 dyes were detected using 580–630 nm and 650–720 filters, respectively. Depending on the experiment, pixel size was set to 15 nm for membrane sheets and 25 nm for cells. For all images, pinhole size was set to 60 μm. For the imaging of membrane sheets, the focal plane was adjusted slightly above and below the membrane sheet in order to check its two-dimensional structure briefly before imaging. This is possible because membrane sheets become immediately unfocussed when the focal plane is displaced in the z-axis, whereas cells or incompletely generated membrane sheets still show structural elements (*Figure 4— figure supplement 1*).

## STED microscopy image analysis

In imageJ, regions of interest (ROIs) were selected in the far-red channel (Alexa-647; displaying CD151), and propagated to the other respective channel. The Pearson correlation coefficient (PCC) between two channels was calculated with a custom ImageJ macro. For analyzing maxima parameters (size, distances, intensity), a custom ImageJ macro was used. The macro is based on the ImageJ function 'Find Maxima' which is used to detect clusters or viral particles and gives their position in pixel coordinates. Prior to maxima detection, images were smoothed with a Gaussian blur ($\sigma = 1$) to reduce false positive maxima due to noise. Cluster intensities were calculated by centering five pixel diameter circular ROIs onto the cluster positions. Cluster intensities were averaged for each cell. The exact cluster and PsV positions were determined by calculating the center of mass of fluorescence within the circular ROIs and used for further analysis. Maxima with low mean intensity (< 2 - 5 a.u., depending on the experiment and channel) were excluded. Intensity measurements were corrected for local background measured next to the cell. For each PsV, the distances to the CD151 clusters were calculated, and the shortest distance was determined. In some cases, we found only a few PsVs per cell. Therefore, all PsV distances were averaged for one condition from one biological replicate (roughly 20 cells). The cluster size was determined by applying a horizontal and a vertical 3 × 31 pixels line scan centered at the cluster position. The linescans were fitted to a Gaussian yielding the full width at half maximum, which corresponds to the cluster size. Depending on the fit quality, either the value from the horizontal or vertical line scan was used. Clusters that have both fits with a R-square $R^2 < 0.8$ (centered peak) were excluded. The cluster size values for each membrane sheet were averaged. Measurements of the CD151 level were performed in the same ROIs as used for the cluster size analysis. Intensity measurements were corrected for local background measured next to the membrane sheets.

## Tracking of PsVs and CD151-CFP

HPV16 PsVs and CD151-CFP accumulations were tracked using the 'Find Maxima' function of ImageJ. For determining the exact object position, a circular ROI with a diameter of 5 pixels was centered on the detected maxima and the center of mass of the fluorescence within the ROI was measured. The objects were tracked over a period of maximally 180 s with a time interval of 6 s between frames, yielding x- and y-positions of the objects over time. The mean-square displacement (MSD) was calculated like described previously (*Lang et al., 2000*) and plotted against the time intervals. For CD151 MSD versus time plot indicated unidirectional flow, which was ignored when determining the diffusion coefficient from a fitted linear regression line (D). The coefficient was calculated for each single track and an average diffusion coefficient was calculated. Five movies

recorded by TIRF microscopy, published and described in Scheffer et al., were analyzed (for details of TIRF microscopy see *Scheffer et al., 2013*).

## Statistics

Data analysis was performed using Statistical Software R (2017, version 3.3.3) from R Core Team (R: A language and environment for statistical computing. R Foundation for Statistical Computing, Vienna, Austria). All values from two conditions that were compared in each statistical assay were tested for normality using Shapiro-Wilk test. If the values followed normal distribution (p>0.05), they were further tested for homogeneity of variances using Bartlett, Fligner-Killeen and Levene's test. Samples were considered as homoscedastic if all three tests displayed p>0.05, and as heteroscedastic if one of them displayed p≤0.05. Differences between the groups for homoscedastic samples were analyzed using two sample t-test and for heteroscedastic with Welch two sample t-test. In the case that values did not follow normal distribution (p≤0.05), Wilcoxon rank sum test was applied. Exact p-values are given for each statistical test where the specific sample (stated in brackets in the Fig. legend if more statistical assays were applied on the same blot or graph) was compared to the control. The 'n' for each presented analysis (stated in Fig. legend) denotes the sample size. Differences between the groups were considered statistically significant when p≤0.05 with the statistical significance marked in the graph (p≤0.05 *, p≤0.01 **, p≤0.001 ***, ns = not significant). All experiments were repeated independently at least three times.

## Acknowledgements

We thank Björn Ahrens, Kirsten Freitag, Konstanze Scheffer, and Anna-Lena Loster for technical support, Dr. Niels Lemmermann for his comments on the manuscript. Snježana Mikuličić, was supported by a fellowship of the German Academic Exchange Service (Deutscher akademischer Austauschdienst, DAAD). Luise Florin and Thorsten Lang were supported by grants from the German Science Foundation (Deutsche Forschungsgemeinschaft, DFG; FL 696/3–1, LA 1272/8–1). Karina Reiss was supported by the DFG, CRC877 (A4) and the Cluster of Excellence 'Inflammation at Interfaces'.

## Additional information

### Funding

| Funder | Grant reference number | Author |
|---|---|---|
| Deutscher Akademischer Austauschdienst | | Snježana Mikuličić |
| Deutsche Forschungsgemeinschaft | CRC877 (A4) | Karina Reiss |
| Deutsche Forschungsgemeinschaft | LA 1272/8-1, FL 696/3-1 | Thorsten Lang Luise Florin |

The funders had no role in study design, data collection and interpretation, or the decision to submit the work for publication.

### Author contributions

Snježana Mikuličić, Conceptualization, Data curation, Formal analysis, Funding acquisition, Validation, Investigation, Visualization, Writing—original draft, Writing—review and editing; Jérôme Finke, Data curation, Formal analysis, Validation, Visualization, Writing—review and editing; Fatima Boukhallouk, Data curation, Validation; Elena Wüstenhagen, Formal analysis, Investigation, Visualization, Writing—review and editing; Dominik Sons, Data curation, Writing—review and editing; Yahya Homsi, Conceptualization, Writing—review and editing; Karina Reiss, Conceptualization, Resources, Funding acquisition, Writing—review and editing; Thorsten Lang, Conceptualization, Formal analysis, Supervision, Funding acquisition, Writing—original draft, Project administration, Writing—review and editing; Luise Florin, Conceptualization, Resources, Supervision, Funding acquisition, Investigation, Writing—original draft, Project administration, Writing—review and editing

Author ORCIDs
Elena Wüstenhagen (iD) http://orcid.org/0000-0002-5420-6536
Thorsten Lang (iD) http://orcid.org/0000-0002-9128-0137
Luise Florin (iD) https://orcid.org/0000-0003-4310-7329

Decision letter and Author response
Decision letter https://doi.org/10.7554/eLife.44345.020
Author response https://doi.org/10.7554/eLife.44345.021

## Additional files

### Supplementary files
• Transparent reporting form
DOI: https://doi.org/10.7554/eLife.44345.018

### Data availability
All data generated or analysed during this study are included in the manuscript and supporting files.

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
