## [Decision Letter]

Thank you for submitting your article "ADAM17-mediated signaling is required for virus entry platform assembly" for consideration by *eLife*. Your article has been reviewed by three peer reviewers, and the evaluation has been overseen by a Reviewing Editor and Anna Akhmanova as the Senior Editor. The following individuals involved in review of your submission have agreed to reveal their identity: Baki Akgül (Reviewer #1) and Gisa Gerold (Reviewer #2).

The reviewers have discussed the reviews with one another and the Reviewing Editor has drafted this decision to help you prepare a revised submission.

Summary:

In their interesting manuscript "ADAM17-mediated signaling is required for virus entry platform assembly," the authors study the molecular events associated with infectious cellular internalization of Human Papillomavirus (HPV) by keratinocytes. They use a rich combination of molecular biology and fluorescence microscopy techniques to show that the proteinase ADAM17 activates signaling that triggers the formation of the endocytic platform to which the virus binds. This secondary entry receptor complex is a cluster of proteins, among which include the tetraspanin CD151 and the epidermal growth factor receptor (EGFR). The arguments are convincing and this work clearly represents an important step to clarify the issue under study. Overall, this paper provides novel mechanistic insights into how the HPV entry platform is formed, and the data are mostly convincing and robust. We believe that addressing the following major concerns would greatly improve the paper.

Essential revisions:

1) "Hence, virus contact to the cellular surface does not activate the signaling pathway for which ADAM17 is required." Figure 6A shows a western blot for contr.si cells treated with pseudovirus. Here, phosphorylation of ERK1/2 can be seen. Therefore, in order to show that the virus does not lead to p-ERK1/2, a blot with extracts from contr.si cells without virus treatment needs to be included. Please also include the information in the figure, whether the extracts are from virus-infected or from non-infected cells.

2) Figure 4: An internalization assay would be needed to prove reduced internalization. If such an assay is technically challenging, the percentage of L1 at the surface and in intracellular compartments at different time points post temperature shift should be quantified.

3) Figure 5: To exclude that modified virions are transferred between donor and recipient cells, it would be beneficial to purify the virions released from the donor cells and add purified virions onto naïve cells.

4) Figure 6: A clear link between ADAM17 and EGF is missing. The fact that the KD phenotype can be reversed (under serum starvation conditions) by exogenous EGF, only implies that EGF can enhance uptake. Here the control siRNA treatment +/- EGF is missing to clarify whether EGF also enhances HPV entry in the presence of ADAM17. A clear link could only be shown using a combination of active site mutant ADAM17, EGF (or other ADAM17 substrate) quantification in the extracellular milieu, EGF neutralization, and EGFR inhibition assays.

5) Figure 6: ADAM17 processes a variety of membrane-bound precursors of growth factors and cytokines, e.g., TNF or Notch, but also integral membrane proteins, e.g., TNF receptor or growth hormone receptor; thus the link to EGF shedding should be more clearly worked out.

6) Figure 6: Which of the many EGFR ligands or other growth factor receptor ligands is involved?

7) Figure 6: EGFR signaling is regulating expression of multiple genes. 48 h post siRNA transfection expression profiles of several proteins critical for HPV entry may have changed. Additional experiments with short-term inhibitor treatment could solve this problem.

8) The whole virus/platform colocalization process is long and takes several hours. A question that is not fully tackled by the authors is the explanation for this duration. Can the proposed model of molecular cascade provide explanations, at least hints? Could it be quantified in terms of diffusion coefficients of the molecular partners? This would strengthen the model by making it realistic from a quantitative perspective.

9) One of the strengths of this work is the use of modern super-resolution microscopy (dual color STED) to go beyond the diffraction limit. In this work, it is notably used to measure CD151 cluster sizes. I also understand that CD151-L1 colocalization quantification was performed by STED. Why hasn't the same approach used to quantify the colocalization of CD151 and EGFR, in order to strengthen the conclusions of Figure 3? The same remark holds for Figure 2.

---

## [Author Response]

Essential revisions:1) "Hence, virus contact to the cellular surface does not activate the signaling pathway for which ADAM17 is required." Figure 6A shows a western blot for contr.si cells treated with pseudovirus. Here, phosphorylation of ERK1/2 can be seen. Therefore, in order to show that the virus does not lead to p-ERK1/2, a blot with extracts from contr.si cells without virus treatment needs to be included. Please also include the information in the figure, whether the extracts are from virus-infected or from non-infected cells.

Control si cells with and without PsV were shown in the original manuscript in the Figure 6B (+PsV) and Figure 6—figure supplement 1A (-PsV) (now labeled in the figure). No difference was observed. To be more specific, we now investigated in detail the effect of PsV preparation on ERK phosphorylation in non-treated cells. The finding is now shown in Figure 6—figure supplement 1C. A mock preparation was used as control. During mock preparation (M), cells not-expressing HPV16 capsid proteins undergo the same procedure as cells expressing HPV capsid proteins for PsVs preparation (V). On the one hand, no difference in ERK activation was observed after mock or PsVs treatment supporting our statement that “virus contact to the cellular surface does not activate the signaling pathway for which ADAM17 is required”. On the other hand, we detected a difference between ERK levels of non-treated and treated cells independent of the presence of PsVs which indicates that in our PsVs preparation we co-purify ERK activating molecules which are also present in the mock control. For these reasons a true non-PsV negative is not possible. Therefore, we now removed our earlier statement, which however is not relevant for our main conclusion.

2) Figure 4: An internalization assay would be needed to prove reduced internalization. If such an assay is technically challenging, the percentage of L1 at the surface and in intracellular compartments at different time points post temperature shift should be quantified.

An HPV internalization assay (as published previously by Milne 2005, Wüstenhagen, 2016) is now included as Figure 2—figure supplement 4C and D. Internalized virus particles are protected from degradation by proteinase K. Cytochalasin D blocks virus internalization and enables proteinase K-mediated degradation of extracellular viral L1 capsid protein. By contrast, L1 in control or ADAM17 siRNA treated cells showed comparable L1 protection. Hence, 24h post infection, comparable amounts of PsVs are internalized. Still, ADAM17 depletion reduces infection and L1-cleavage.

These observations suggest that although ADAM17 depletion reduces viral particles entry into the infectious pathway, it concomitantly increases PsVs internalization into a non-infectious pathway. This argument has been added “Investigating the cleavage pattern of the viral L1 protein (L1 priming), we find that L1 is neither a substrate for ADAM17 (Figure 2—figure supplement 4A and B) nor internalization of viral particles is blocked by ADAM17 depletion (Figure 2—figure supplement 4C-D). Instead, L1 cleavage products substantially decreased in ADAM17-depleted cells (Figure 2—figure supplement 4E and F). Therefore, we speculate that ADAM17 activity is required for viral particle association with L1-priming proteases such as kallikrein-8 prior to endocytosis and/or in intracellular compartments of the infectious entry pathway (47). Inhibition of these early steps on the plasma membrane rather leads to virus internalization into a non-infectious entry pathway as described earlier (48).”

3) Figure 5: To exclude that modified virions are transferred between donor and recipient cells, it would be beneficial to purify the virions released from the donor cells and add purified virions onto naïve cells.

We actually do not exclude but assume that modified virions are transferred. To avoid any confusion we have added to the Discussion “In the co-culture transwell assay modified viral particles are transferred only from the donor cells to the recipient cells”.

We also discuss the possibility that released PsVs are decorated with different amounts of GFs: “This mutual influence is explained by an exchange of bioactive molecules between donor and recipient cells in both directions. The association of bioactive molecules with the viral capsid as proposed by the Ozbun group (Surviladze, Dziduszko and Ozbun, 2012) seems to increase the effect.”

4) Figure 6: A clear link between ADAM17 and EGF is missing. The fact that the KD phenotype can be reversed (under serum starvation conditions) by exogenous EGF, only implies that EGF can enhance uptake. Here the control siRNA treatment +/- EGF is missing to clarify whether EGF also enhances HPV entry in the presence of ADAM17. A clear link could only be shown using a combination of active site mutant ADAM17, EGF (or other ADAM17 substrate) quantification in the extracellular milieu, EGF neutralization, and EGFR inhibition assays.

We thank the reviewers for raising this important issue. EGF is only one out of seven known EGFR ligands. Therefore, the EGF effects only point towards the EGFR being involved in signaling, but not necessarily towards EGF as a ligand. Actually, EGF is an unlikely candidate, because its production depends more on shedding of other ADAM family members. The functional importance of EGFR is now shown as new Figure 6A by performing EGFR inhibition assay.

Candidates for GFs depending on ADAM17 sheddase activity include HB-EGF, TGF-α, and AREG. We now show reduced levels of AREG and TGF-α in the supernatant of ADAM17 depleted cell, confirming that also in our system the production of AREG and TGF-α depends on ADAM17. We also analyzed HB-EGF, but were not able to detect it in the western blot, most likely because after cleavage it does not diffuse into the supernatant but remains immediately bound to HSPGs.

AREG and TGF-α play a role in infection because infectivity is reduced after adding neutralizing antibodies. Finally, adding AREG, TGF-α, and HB-EGF to the supernatant of serum-starved or ADAM17-depleted cells increases ERK phosphorylation and infection.

Together, the new data incorporated as Figure 7 and Figure 7—figure supplement 1 provide a link between ADAM17-mediated GFs shedding, activation of the ERK1/2 signaling pathway and HPV16 infection. We added a paragraph with a detailed discussion to “ADAM17-released growth factors”.

We would like to thank again the reviewers for pointing out this important issue, which helped us to improve the manuscript.

5) Figure 6: ADAM17 processes a variety of membrane-bound precursors of growth factors and cytokines, e.g., TNF or Notch, but also integral membrane proteins, e.g., TNF receptor or growth hormone receptor; thus the link to EGF shedding should be more clearly worked out.

The link between ADAM17 and GFs is now shown in new Figure 7 and Figure 7—figure supplement 1. Please see our detailed reply above.

However, we did not find a specific GF, which is not expected, as the EGFR has many ligands. We also cannot exclude the involvement of further ligands. We state in the Discussion: “The effect of EGFR, AREG or TGF-α inhibition is much lower than that of the ADAM17 siRNA, therefore, we have to assume that ADAM17 not only activates the EGFR but also activates additional

GFRs.” and “Due to the plethora of ADAM17 substrates (Gooz, 2010; Reiss and Saftig, 2009; Edwards, Handsley and Pennington, 2008) and the complex crosstalk between signaling pathways, also, ADAM17-dependent shedding of additional membrane-bound precursors and cleavage of integral membrane proteins might be involved in this scenario.”

6) Figure 6: Which of the many EGFR ligands or other growth factor receptor ligands is involved?

The involvement of AREG, TGF-α, and HB-EGF is now uncovered. Please see our reply to the previous point.

7) Figure 6: EGFR signaling is regulating expression of multiple genes. 48 h post siRNA transfection expression profiles of several proteins critical for HPV entry may have changed. Additional experiments with short-term inhibitor treatment could solve this problem.

We now performed experiments using Cetuximab, a blocking-antibody of EGFR, which we incubated for only 1h prior to PsVs addition. (Results are shown in as new Figure 6A). Effect of ADAM17 short-term inhibitors TAPI-0 and GW is shown in Figure 1—figure supplement 1A.

8) The whole virus/platform colocalization process is long and takes several hours. A question that is not fully tackled by the authors is the explanation for this duration. Can the proposed model of molecular cascade provide explanations, at least hints? Could it be quantified in terms of diffusion coefficients of the molecular partners? This would strengthen the model by making it realistic from a quantitative perspective.

We analyzed the mobility of viral particles. They are essentially immobile (Figure 2—figure supplement 2) as intensively investigated and discussed by Schelhaas et al., 2008. This is in contrast to tetraspanins, known to move in the cell membrane (please note that as well we tracked CD151 structures but the pattern in the TIRF field was not well resolved).

In response to the next point, we analyzed by superresolution microscopy the overlap between CD151 and L1, which increases over time when analyzed with conventional microscopy (old Figure 2). Using STED microscopy, we find 30% of the viral particles readily associating with CD151 clusters already 15 min after addition. In the next few hours, this fraction increases to almost 50%, and the CD151 signal at the viral binding sites becomes brighter (new Figure 2 A-C). Both effects explain the increase in overlap seen with conventional microscopy (this data is still in the manuscript as Figure 2—figure supplement 1).

All findings together indicate that platform maturation is associated with the recruitment of CD151 molecules to an essentially immobile, cell surface bound viral particle, which takes a few hours. This is discussed in the subsection “Antibodies and plasmids”.

Surprisingly, nanoscopy uncovered the above-mentioned large fraction of viral particles associated with CD151 clusters already present briefly after binding. It is possible that virus platform formation does not strictly depend on virus unbinding and rebinding, which occurs as documented in the trans-well assay, but may not be essential for infectivity. We have added to the Discussion: “Examining at nanoscale resolution the secondary entry receptor complex component CD151, briefly after binding, we find viral particles associated with CD151 clusters and over the next hours an enrichment of CD151 at the virus binding site.” And modified the current model accordingly. “It is possible that a large fraction of viral particles are not released from the cell surface, as expected from the primary receptor/secondary receptor model, but can remain at or in close proximity to the initial binding site.”

9) One of the strengths of this work is the use of modern super-resolution microscopy (dual color STED) to go beyond the diffraction limit. In this work, it is notably used to measure CD151 cluster sizes. I also understand that CD151-L1 colocalization quantification was performed by STED.

We have not employed STED microscopy in old Figure 2, showing that the overlap between CD151 and L1 increases over time. This experiment has been repeated with STED microscopy and revealed the above-mentioned interesting details, allowing for a better understanding of platform formation (see Figure 2A-C and reply to previous point).

Why hasn't the same approach used to quantify the colocalization of CD151 and EGFR, in order to strengthen the conclusions of Figure 3?

As requested, we have analyzed the colocalization between CD151 and EGFR by STED microscopy. We confirm colocalization see with conventional microscopy, which however is not dependent on ADAM17 (See new Figure 3C and D).

The same remark holds for Figure 2.

Has been done (see previous point).